# Geometric deep learning of protein–DNA binding specificity

Raktim Mitra ●[1], Jinsen Li ●[1], Jared M. Sagendorf[1,7], Yibei Jiang ●[1], Ari S. Cohen[1], Tsu-Pei Chiu ●[1], Cameron J. Glasscock[2,3] & Remo Rohs ●[1,4,5,6] ✉

Predicting protein–DNA binding specificity is a challenging yet essential task for understanding gene regulation. Protein–DNA complexes usually exhibit binding to a selected DNA target site, whereas a protein binds, with varying degrees of binding specificity, to a wide range of DNA sequences. This information is not directly accessible in a single structure. Here, to access this information, we present Deep Predictor of Binding Specificity (DeepPBS), a geometric deep-learning model designed to predict binding specificity from protein–DNA structure. DeepPBS can be applied to experimental or predicted structures. Interpretable protein heavy atom importance scores for interface residues can be extracted. When aggregated at the protein residue level, these scores are validated through mutagenesis experiments. Applied to designed proteins targeting specific DNA sequences, DeepPBS was demonstrated to predict experimentally measured binding specificity. DeepPBS offers a foundation for machine-aided studies that advance our understanding of molecular interactions and guide experimental designs and synthetic biology.

Transcription factors play critical roles in various regulatory functions that are essential to all aspects of life[1]. Therefore, understanding the mechanisms by which proteins target specific DNA sequences is crucial[2]. Extensive research has uncovered myriad binding mechanisms that lead to specific high-affinity binding, including strong electrostatic interaction of arginine residues in the DNA minor groove[3], deoxyribose sugar-phenylalanine stacking[4], bidentate hydrogen bonds (H-bonds) between guanine (G) and arginine (Arg) in the major groove[5], and other interactions[6–8].

Protein–DNA structures are typically[9] obtained through X-ray crystallography, nuclear magnetic resonance spectroscopy or cryo-electron microscopy experiments and stored in the Protein Data Bank (PDB)[10]. Generally, these structures display one bound DNA sequence and the associated physicochemical interactions[6] but do not encompass the full range of potentially bound DNA sequences. Conversely, this information can be experimentally obtained through protein-binding microarray[11], systematic evolution of ligands by exponential enrichment combined with high-throughput sequencing (SELEX–seq)[12], chromatin immunoprecipitation followed by sequencing[13], high-throughput SELEX[14] or related high-throughput approaches[15]. These experiments capture the range of possible bound DNA sequences but do not necessarily provide structural information. In essence, these sets of experiments are complementary, and manual examination is often required to correlate molecular interaction details from structural data with binding specificity data[6].

Predicting binding specificity for a given protein sequence, across protein families, remains a challenging and unsolved problem, despite progress for specific protein families[16–23]. Structural changes in the context of binding, along with large mechanistic diversity, contribute to the difficulty[15,24]. Protein–DNA structures contain valuable information that artificial intelligence can leverage to achieve generalizability across protein families. In this framework, we introduce

[1]Department of Quantitative and Computational Biology, University of Southern California, Los Angeles, CA, USA. [2]Department of Biochemistry, University of Washington, Seattle, WA, USA. [3]Institute for Protein Design, University of Washington, Seattle, WA, USA. [4]Department of Chemistry, University of Southern California, Los Angeles, CA, USA. [5]Department of Physics and Astronomy, University of Southern California, Los Angeles, CA, USA. [6]Thomas Lord Department of Computer Science, University of Southern California, Los Angeles, CA, USA. [7]Present address: Department of Bioengineering and Therapeutic Sciences, University of California San Francisco, San Francisco, CA, USA. ✉e-mail: rohs@usc.edu

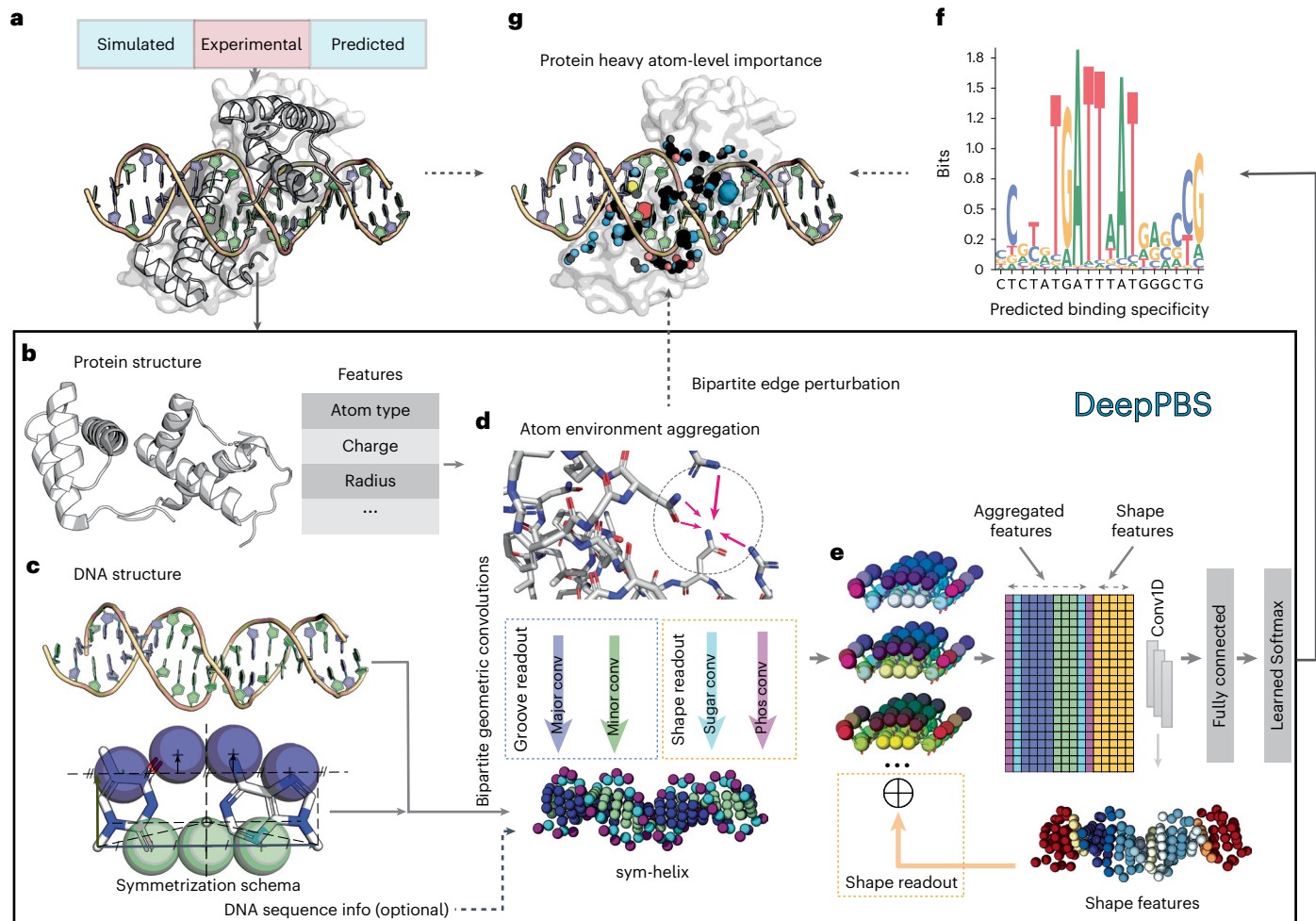

**Fig. 1 | Schematic illustration of the DeepPBS framework. a**, DeepPBS input (PDB ID 2R5Y in this example) and possible input sources. **b**, Protein structure (heavy atom graph, with features computed for each vertex). **c**, Symmetrization schema in base-pair frame applied to DNA structure, resulting in a sym-helix. **d**, Spatial graph convolution on the protein graph for atom environment aggregation, followed by bipartite geometric convolutions from protein graph vertices to sym-helix points (shown as spheres with specific colors for major groove, minor groove, phosphate and sugar). **e**, Three-dimensional sym-helix is flattened with aggregated information (concatenated with computed shape features) into a 1D representation, followed by 1D convolutions and regression onto base pair probabilities. **f**, DeepPBS outputs binding specificity. **g**, Effect of perturbing bipartite edges involved in **d** can be measured in terms of changes in the output, providing an effective measure of interpretability. Phos, phosphate; conv, convolutions.

Deep Predictor of Binding Specificity (DeepPBS). This deep-learning model is designed to capture the physicochemical and geometric contexts of protein–DNA interactions to predict binding specificity, represented as a position weight matrix (PWM)[25] based on a given protein–DNA structure (Fig. 1a). DeepPBS functions across protein families (Fig. 2) and acts as a bridge between structure-determining and binding specificity-determining experiments.

Input of DeepPBS is not limited to experimental structures (Fig. 1a). The rapid advancement of protein structure prediction methods, including AlphaFold[26], OpenFold[27] and RoseTTAFold[28], along with protein–DNA complex modelers, such as RoseTTAFoldNA (RFNA)[29], RoseTTAFold All-Atom[30], MELD-DNA[31] and AlphaFold3 (ref. 32), have led to an exponential increase in the availability of structural data for analysis. This scenario highlights the growing need for a generalized computational model to analyze protein–DNA structures. We demonstrate how DeepPBS can work in conjunction with structure prediction methods for predicting specificity for proteins without available experimental structures (Fig. 3a–d). In addition, the design of a protein–DNA complex can be improved by optimizing bound DNA using DeepPBS feedback (Fig. 3e–g). We show that this pipeline is competitive with the recent family-specific model rCLAMPS[17] (Fig. 3h,i) while being more

generalizable: specifically, DeepPBS is protein family-agnostic, can handle biological assemblies and can predict DNA flanking preferences.

In terms of interpretability, 'relative importance' (RI) scores for different heavy atoms in proteins that are involved in interactions with DNA can be extracted from DeepPBS (Fig. 4). As a case study on an important protein for cancer development, we analyze the p53–DNA interface via these RI scores and relate them with existing literature for validation. Additionally, we show that the DeepPBS scores align well with existing knowledge and can be aggregated to produce reasonable agreement with alanine scanning mutagenesis experiments[33] (Fig. 4h).

In additional proof-of-principle studies, we apply DeepPBS to in silico-designed protein–DNA complexes targeting specific DNA sequences (Fig. 5), obtained from a recent study that combines structural design with DNA mutagenesis experiments[34]. Finally, we show that DeepPBS can also be used to analyze molecular simulation trajectories. We demonstrate an example by applying DeepPBS to a molecular dynamics (MD) simulation of Extradenticle (Exd) and Sex combs reduced (Scr) Hox heterodimer in complex with DNA[35] with an AlphaFold-based modeled protein linker (Supplementary Section 10, Supplementary Fig. 6 and Supplementary Video 1). DeepPBS is available as a webserver at https://deeppbs.usc.edu.

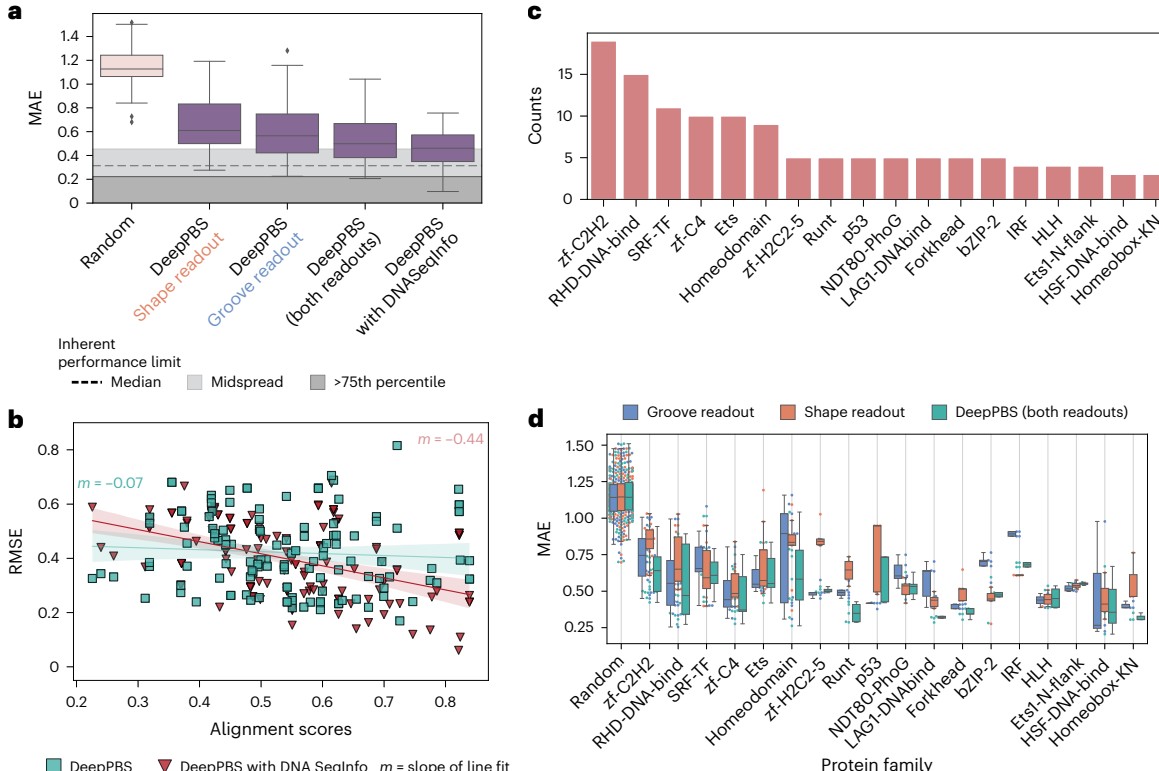

**Fig. 2 | Performance of DeepPBS for predicting binding specificity across protein families for experimentally determined structures. a**, Prediction performances of DeepPBS along with 'groove readout', 'shape readout' and 'with DNA SeqInfo' variations, on benchmark set (biological assemblies corresponding to $n = 130$ protein chains (for each box plot); Supplementary Section 1). MAE, mean absolute error; RMSE, root mean squared error. **b**, Performances of DeepPBS and 'with DNA SeqInfo' models in context of PWM–co-crystal-derived DNA alignment score (Supplementary Section 2). The shaded regions indicate the 95% confidence interval for the corresponding linear fit. The MAE equivalent of this plot is available as Supplementary Fig. 12, showing similar trends. **c**,

Abundances of various protein families (as appearing in PFAM annotations) in constructed benchmark set (counts >3). **d**, Performances of DeepPBS, groove readout and shape readout models across various protein families (counts >3) (biological assemblies corresponding to $n$ protein chains (for each family), where $n$ is as described in **c**, total unique $n = 130$). All benchmark predictions are made by an ensemble average of five models trained via cross-validation. Cross-validation performances of individual trained models are shown in Supplementary Fig. 5a. For the box plots in **a** and **d**, the lower limit represents the lower quartile, the middle line represents the median and the upper limit represents the upper quartile.

## Results

### The DeepPBS framework

The DeepPBS framework is illustrated in Fig. 1. Input to DeepPBS (Fig. 1a) is composed of one protein–DNA complex structure, with one or more protein chains bound to a DNA double helix. Potential sources for such structures include experimental data (for example, PDB[10]), molecular simulation snapshots or designed complexes. DeepPBS processes the structure as a bipartite graph with distinct spatial graph representations for protein and DNA components. The protein graph is an atom-based graph, with heavy atoms as vertices. Several features are computed on these vertices (Fig. 1b). Further information on protein representation and feature computation is available in Supplementary Section 4. We represent DNA as a symmetrized helix (sym-helix), as detailed in Methods. This representation removes any sequence identity that the DNA possesses, while preserving the shape of the double helix[3]. Optionally, DNA sequence information can be reintroduced as a feature on the sym-helix points.

DeepPBS performs a series of spatial graph convolutions on the protein graph to aggregate atomic neighborhood information (Fig. 1d). The next crucial component of DeepPBS consists of a set of bipartite geometric convolutions applied from the protein graph to the sym-helix (Fig. 1d). Specific chemical interactions (for example, hydrogen bonds) depend on both location and orientation[5]. DeepPBS learns how the geometric orientation of the sym-helix points is associated with the orientations and chemistry of neighboring protein residues. Four distinct bipartite convolutions are employed for the sym-helix points,

corresponding to the major groove, the minor groove and the phosphate and sugar moieties. Major and minor groove convolutions are referred to as 'groove readout'. This term was chosen over the term 'base readout' due to the removal of base identity in the sym-helix. Phosphate and sugar moiety convolutions, combined with DNA shape information, form the 'shape readout' (Fig. 1e). The 'groove readout' and 'shape readout' factors collaboratively determine binding specificity to varying extents for different protein families. At this point, the sym-helix representation enables a straightforward flattening of aggregated features on the three-dimensional sym-helix to the one-dimensional (1D) base pair-level features. By adding DNA shape information and implementing 1D convolutional neural network and prediction layers (Fig. 1e), DeepPBS ultimately predicts binding specificity (Fig. 1f). Further architectural details are described in Supplementary Section 5.

Lack of an existing published standard dataset for predicting binding specificity across protein families from protein–DNA complex structure data made it necessary for us to build a dataset for cross-validation and benchmarking. Details of this process can be found in Methods.

### DeepPBS performance for experimentally determined structures

The DeepPBS ensemble (Methods) was employed to evaluate model performance against a benchmark set, as outlined in Supplementary Section 1. The DeepPBS architecture allows models to be trained on two mechanisms: 'groove readout', which does not involve backbone

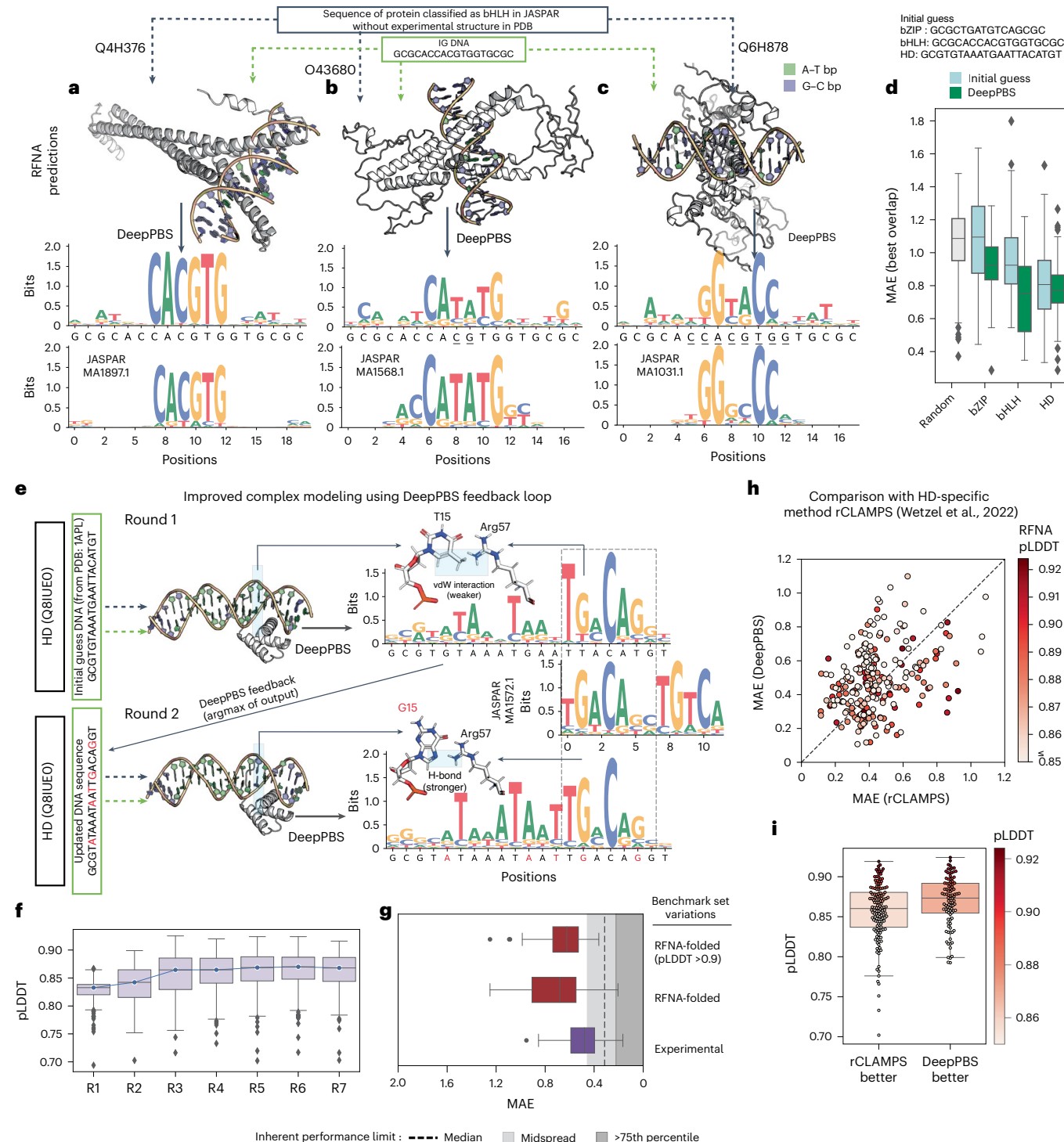

**Fig. 3 | Application of DeepPBS on predicted protein–DNA complex structures.** Various predictive approaches (for example, RFNA and MELD-DNA) can be used to predict protein–DNA complex structures in the absence of experimental data. DeepPBS can predict binding specificity on the basis of this predicted complex. **a**–**c**, Examples for three full-length bHLH protein sequences: Max homodimer from *Ciona intestinalis* (**a**), TCF21 dimer from *Homo sapiens* (**b**) and OJ1581_H09.2 dimer from *Oryza sativa* (**c**). **d**, Performance of DeepPBS via the same process applied for three different families, bZIP (*n* = 50 predicted assemblies), bHLH (*n* = 49 predicted assemblies) and HD (*n* = 236 predicted assemblies), compared with baselines determined for random (drawn from uniform) and IG DNA sequences. Each protein has a unique JASPAR annotation and lacks an experimental structure for the complex. Structures for protein complexes were predicted by RFNA. Proteins passed the preprocessing criterion of DeepPBS. **e**, One iteration of DeepPBS feedback, demonstrated for human

TGIF2LY protein. vdW, van der Waals. **f**, RFNA-predicted LDDT[44] score over rounds 1–7 of DeepPBS feedback loop (*n* = 236 predicted assemblies). **g**, Comparison of DeepPBS ensemble performance on benchmark set for experimental and RFNA folded structures (for all processable RFNA-folded structures with greater than 500 contact counts (5 Å cutoff) to the DNA helix (*n* = 98 predicted assemblies) and high confidence (pLDDT >0.9) set (*n* = 31 predicted assemblies)). **h**, Comparison of DeepPBS predictions against HD family-specific method rCLAMPS, color-coded by pLDDT. Diagonal dashed line represents *y* = *x*. **i**, Distribution of pLDDT for two cases: when DeepPBS outperforms rCLAMPS (below diagonal in **h**) and vice versa (above diagonal in **h**) (*n* = 140 (left) and 96 (right) predicted assemblies). The box colors denote the average pLDDT, using the same colormap as in **h**. For the box plots in **d**, **f**, **g** and **i**, the lower limit represents lower quartile, the center line represents the median and the upper limit represents the upper quartile. The whiskers do not include outliers.

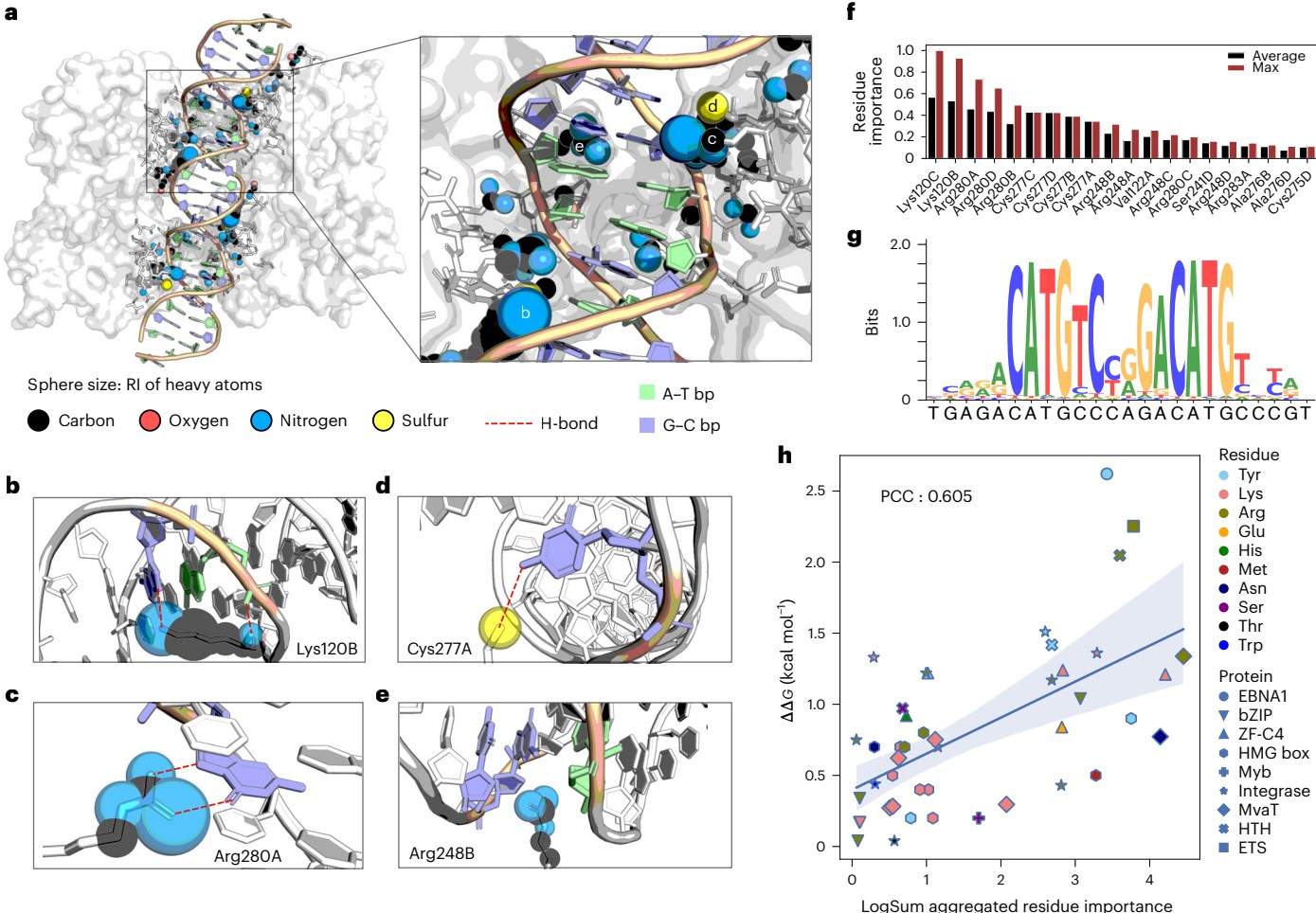

**Fig. 4 | Visualization of DeepPBS importance scores in p53–DNA interface as a case study, and experimental validation.** p53 binds to DNA as a tetramer with two symmetric protein–DNA interfaces[47] (A, B, C and D refer to each monomer; PDB ID: 3Q05). **a**, Relative importance (RI) score (normalized by maximum across atoms) calculated for heavy atoms (denoted by sphere sizes: largest 1, smallest 0) within 5 Å of the sym-helix. **b–e**, Zoomed-in view of specific interactions by protein–DNA interface residues Lys120B (**b**), Arg280A (**c**), Cys277A

(**d**) and Arg248B (**e**) with RI scores assigned by DeepPBS. **f**, Residue importance computed by average and maximum aggregation of heavy atom importance (top 20). **g**, DeepPBS prediction. **h**, Comparison of log sum aggregated residue importance computed from DeepPBS ensemble, with experimental free energy change (ΔΔ*G*) determined by alanine scanning mutagenesis experiments. The blue line indicates linear regression fit. The light-blue region indicates the corresponding 95% confidence interval computed via bootstrapping mean.

convolutions and excludes shape information, and 'shape readout', which does not involve groove convolutions (Fig. 1d,e). Benchmark performances of DeepPBS (which performs both 'groove readout' and 'shape readout' modes combined) and these two variations are shown in Fig. 2a. The 'groove readout' version does better than the 'shape readout' version in terms of median performance, while the DeepPBS model improves upon either component in isolation (two-sided *t*-test *P* value <0.01; Fig. 2a). Pairwise *t*-test *P* values for these variations are available (Supplementary Data 1). A discussion of the outliers in Fig. 2a is provided in Supplementary Section 12.

The dataset was constructed using experimentally determined structures; thus, the co-crystal structure-derived DNA sequence typically serves as a reasonable example of a bound sequence. As expected, integrating sequence information into the sym-helix points ('DeepPBS with DNA SeqInfo') enhanced performance (Fig. 2a), significantly closing the gap toward the inherent performance limit in the dataset. The inherent performance limit originates from the fact that for the same protein the binding specificity data presented by two databases[36,37] used to create the dataset may disagree to some extent (Supplementary Fig. 1c). We computed the distribution of disagreement across all unique PWMs appearing in both databases (Supplementary Section 1). However, from both interpretability and design perspectives,

particularly when the bound DNA sequence may not be representative, the 'DeepPBS' model is optimal due to its low sensitivity to the DNA sequence in the structure. This fact is evidenced by comparing performances of the 'DeepPBS' and 'DeepPBS with DNA SeqInfo' models in the context of the PWM–co-crystal-derived DNA alignment score (Supplementary Section 1). Compared with the line fit to the variation with DNA sequence information (slope −0.44 for root mean squared error (RMSE), slope −0.62 for mean absolute error (MAE); Supplementary Fig. 11), the slope of the line fit to the DeepPBS predictions was closer to zero (Fig. 2b and Supplementary Fig. 11).

As an example, we show the DeepPBS ensemble prediction for the NF-κB biological assembly from the benchmark dataset. Although the co-crystal structure-derived DNA sequence was not of the highest binding affinity, as indicated by experimental data from HOCOMOCO[37], our prediction circumvented this issue, predicting a binding specificity that was more closely aligned with the experimental data (Supplementary Fig. 5d). Similar trends (Supplementary Fig. 5a–c) can be observed from cross-validation predictions by individual DeepPBS models (Methods). We also included example DeepPBS ensemble predictions (Supplementary Fig. 7) for structures in the PDB that correspond to specific interactions but do not have a PWM in the two binding specificity databases considered (Methods). In addition, example DeepPBS ensemble

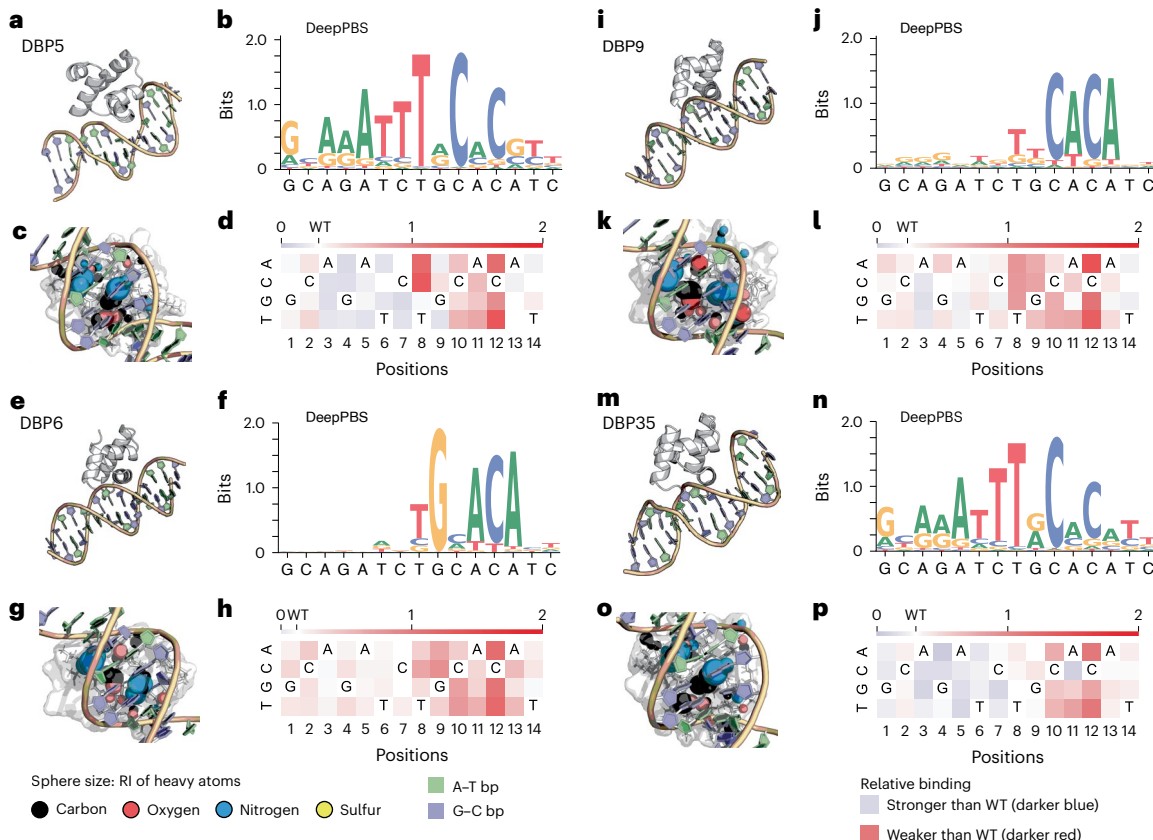

**Fig. 5 | Application of DeepPBS to in silico-designed HTH scaffolds targeting a specific DNA sequence. a,e,i,m**, Design models of four different synthetic HTH proteins targeting the DNA sequence GCAGATCTGCACATC (design based on DNA sequence from PDB ID 1L3L, canonical B-DNA structure used for **e** and **i**, co-crystal-derived DNA structure used for **a** and **m**), obtained from a recent sequence-specific DNA binder design study[34]. **b,f,j,n**, DeepPBS ensemble predictions based on each design model shown in **a**, **e**, **i** and **m**, respectively. As expected, the predictions for DBP5 and DBP35 were very similar due to comparable designs (see 'Data availability' section). **c,g,k,o**, DeepPBS assessment

of heavy atom level RI scores for each interface in the design models shown in **a**, **e**, **i** and **m**, respectively. **d,h,l,p**, Relative binding activity (phycoerythrin/ fluorescein isothiocyanate normalized to the no-competitor condition) of all possible single base-pair mutations obtained via flow cytometry analysis[34] in yeast display competition assays for each of the four HTH proteins shown in **a**, **e**, **i** and **m**, respectively. Blue indicates competitor mutations where competition was stronger than with the WT competitor, while red indicates competitor mutations where competition was weaker.

predictions (Supplementary Fig. 8) for structures of nonspecific protein–DNA binding (for example, SSO7D–DNA interaction[38]) present in the PDB are presented. These predictions have notably lower information content compared with those in Supplementary Fig. 7.

### DeepPBS captures patterns of family-specific binding modes
Abundances of different protein families in the benchmark set are described in Fig. 2c (Supplementary Fig. 5b for cross-validation set). Family annotations were obtained from the Database of Protein Families (PFAM)[39]. The dataset encompasses a wide range of DNA-binding protein families. Performance of DeepPBS for various protein families provides several key insights. DeepPBS showed reasonable generalizability across protein families, performing well even for families with relatively fewer structures (Fig. 2d and Supplementary Fig. 5c), such as heat shock factor proteins. This observation suggests that the model is learning the underlying mechanisms of protein–DNA binding rather than overfitting on family-specific patterns.

Further validation is provided by comparing performances of the DeepPBS 'groove readout' and 'shape readout' models (Fig. 2d and Supplementary Fig. 5c). For families like zf-C2H2, zf-C4 the 'shape readout' model did not perform as well as the 'groove readout' model. This result aligns with the common understanding of the binding mechanism of these families. For example, zf-C2H2 uses zinc finger motifs to scan DNA for suitable base interactions, with minimal DNA bending or conformational change[40]. This binding mode makes the zf-C2H2 family a popular

target of protein sequence-based binding specificity prediction and design[16,18,19,23,41]. Conversely, families like interferon-regulatory factor (IRF) proteins (Fig. 2d and Supplementary Fig. 5c) and T-box proteins (Supplementary Fig. 5c) showed higher performances for the 'shape readout' model, consistent with their known binding mechanisms that involve significant conformational changes[4,42]. For families such as homeodomain (HD) and forkhead (Fig. 2d and Supplementary Fig. 5c), the DeepPBS model outperformed both the 'groove readout' and 'shape readout' components. This result suggests that the network captures complex higher-order relationships of these components. Pairwise *P* values for the three readout variations for Fig. 2d and Supplementary Fig. 5c are available in Supplementary Data 1.

### Application to in silico-predicted protein–DNA complexes
The DeepPBS framework is not limited to experimental structures. Recent advances in scalable structural prediction approaches, driven by artificial intelligence[26,28], offer unprecedented potential. Specifically, models like RFNA[29] and MELD-DNA[31] can be used to predict the structures of protein–DNA complexes from sequence. Such prediction algorithms have paved the way for DeepPBS to be applicable to proteins that lack experimental DNA-bound structure data.

We suggest one potential approach for working with predictive structures in DeepPBS. First, we make an initial guess for the DNA (IG DNA) sequence bound to each protein of interest based on the corresponding protein family. Then, we use RFNA to predict the

protein–DNA complex structure, followed by DeepPBS to predict binding specificity. We demonstrate this process (Fig. 3a–c) for three proteins classified as basic helix-loop-helix (bHLH) in JASPAR[36]. In all three cases, the PDB lacked experimental protein–DNA complex structures. The IG DNA (Supplementary Section 8) has an enhancer box motif ('CACGTG') in the center, which is known[43] to be a bHLH family target. The first example (UniProt Q4H376; Fig. 3a) is a Max homodimer, for which DeepPBS predicted a specificity closely mirroring that of the IG DNA. The second example (TCF21 dimer, O43680) was more complicated; the central 'CACGTG' motif in the IG DNA was erroneously assumed, yet DeepPBS successfully predicted the correct motif as 'CATATG' (Fig. 3b). The third example (Fig. 3c, protein OJ1581_H09.2, Q6H878) does not conform to any enhancer box motif. Nevertheless, DeepPBS predicted a binding specificity closely mirroring the experimental data (Fig. 3c).

We ran the DeepPBS pipeline for full-length UniProt protein sequences, each with a unique JASPAR entry and no experimental structure for the complex, across three different families (Supplementary Section 8): bZIP, bHLH and HD families. DeepPBS predictions based on RFNA-predicted structures exhibited an improved MAE (that is, closer to experimental data) compared with the IG DNA baseline (Fig. 3d). An application of DeepPBS to a MELD-DNA-predicted complex of the mouse CREB1 protein is demonstrated in Supplementary Fig. 9b. Thus, DeepPBS can take predicted structures from suboptimal DNA sequences and predict binding specificity close to experimental data.

We next explored whether DeepPBS prediction could be used as feedback (in a loop) to enhance modeling of the protein complex (and, subsequently, improve DeepPBS prediction). We demonstrated this process for the human TGIF2LY protein (UniProt ID Q8IUE0, unstructured region trimmed; Supplementary Section 8) in Fig. 3e. In round 1, we applied RFNA to this protein sequence alongside the IG DNA sequence for the HD family and then used the predicted complexes as input for DeepPBS. For IG DNA position T15 (Fig. 3e, round 1), DeepPBS predicted a strong preference for G. In the round 1 RFNA output, Arg57 and T15 were involved in one hydrogen bond (H-bond) and one van der Waals interaction. These interactions are theoretically weaker than the possible bidentate H-bonds between a G and Arg57. In round 2, we altered the RFNA input by taking the argmax (the most preferred sequence) from the DeepPBS output (Fig. 3e, round 2). The subsequently folded structure reflected a more robust bidentate H-bond interaction between G15 and Arg57, with the DeepPBS prediction more closely aligning with the experimental data (note positions (round 2) A18, G19 and T14, corresponding to positions 4–6 in MA1572.1; Fig. 3e).

We repeated this DeepPBS prediction process for a total of seven rounds, for the set of HD monomer sequences (Supplementary Section 8). The RFNA-predicted confidence metric (predicted local distance difference test (pLDDT), LDDT[44] reflects similarity between the predicted and reference structure for a complex; Supplementary Section 8) improved over these rounds (Fig. 3f). To independently evaluate structure quality, we calculated the molecular mechanics and Poisson–Boltzmann surface area[45] binding energy (Supplementary Section 8). From round 1 to round 3+, the number of stable structures (binding energy <0 kJ mol⁻¹) increased (Supplementary Fig. 9c), while their binding energy distributions shifted toward lower values (Supplementary Fig. 9c). DeepPBS performance improved across the five rounds (Supplementary Fig. 9a). We also refolded the benchmark set datapoints via RFNA (Supplementary Section 8) and compared (for the full processable set ($n$ = 98) and a high-confidence set, pLDDT >0.9, $n$ = 31) the performances with the equivalent performance obtained for the experimental structures (Fig. 3g). There is a drop in performance. We can expect that it will improve when future models for structure prediction become available.

The DeepPBS approach for predicting binding specificity fundamentally differs from that of existing methods, which predict binding specificity solely on the basis of protein sequence information.

As a result, comparisons with existing family-specific methods that operate exclusively on protein sequence are unfeasible. However, in conjunction with a complex structure prediction method, we can start from protein sequence information alone and predict binding specificity using DeepPBS. This process can be compared with the recent HD family-specific method, rCLAMPS[17] (Supplementary Section 8). rCLAMPS can predict core 6-mer binding specificities for monomer HD proteins. A comprehensive overview of performances is shown in Fig. 3h. For different significant portions of the data, DeepPBS and rCLAMPS outperformed each other. DeepPBS outperformed rCLAMPS where the pLDDT scores were higher (Fig. 3i). Thus, the DeepPBS pipeline is comparable to rCLAMPS, while having broader applicability across families and biological assemblies as well as not being limited to predicting the DNA core binding region.

### Assessing protein residue importance at p53–DNA interface

The DeepPBS architecture permits intentional activation or deactivation of specific edges in the bipartite geometric convolution stage (Fig. 1d and Supplementary Fig. 4). Perturbing a set of edges in this manner will alter the network-predicted result. The mean absolute difference between the original and altered prediction can be used (with proper normalization) as a quantification of the impact of the perturbed set of edges in determining binding specificity (Fig. 1g, Supplementary Fig. 4 and Methods).

We present results for perturbing edge sets for individual protein heavy atoms, which can also be aggregated to compute residue-level importance. As an example, we examined the protein–DNA interface of p53 (PDB ID: 3Q05), a protein crucial for regulating cancer development and cell apoptosis[46]. The tumor suppressor p53 binds to DNA as a tetramer with two symmetric protein–DNA interfaces[47,48]. We show the RI scores (with min–max normalization applied) calculated for heavy atoms within 5 Å of the sym-helix (Fig. 4a). Sphere sizes in Fig. 4a denote computed RI scores, with the largest being 1 and smallest 0. Lys120 (ref. 49) is involved in both groove readout (H-bond with G) and shape readout-based binding specificity (H-bond with backbone phosphate) (Fig. 4b). The network deems G-Arg280 (ref. 49) bidentate H-bonds as another strong driver of binding specificity[5] (Fig. 4c). Cys277 confers specificity through its thiol sulfur, accepting an H-bond in the major groove[49] (Fig. 4d). Another important residue according to DeepPBS, Arg248 (ref. 50), is present at the minor groove (Fig. 4e). This decision by the model is primarily based on the orientation of arginine relative to the sym-helix, which is devoid of DNA sequence information. Arg248 is attracted through enhanced negative electrostatic potential due to a narrowing of the minor groove where it binds[47]. Among other residues in Fig. 4f, Ser241 is known[50] to be important for stabilizing Arg248. Ala276 (known for causing apoptosis upon mutation[51]) appears as another driver of specificity. This residue has been shown to be a driver of specificity via van der Waals contacts with the methyl group of T in the major groove[49]. The binding specificity prediction of DeepPBS (Fig. 4g) aligns well with known binding patterns of p53, which follows the form RRRC(A/T)(A/T)GYYY (R denotes purine, and Y denotes pyrimidine). The interactions shown here are deemed[46,52] as significant drivers of p53 binding.

### Comparison of residue-level importance with mutagenesis data

We next asked whether DeepPBS-derived importance scores, which reflect the degree to which an interaction determines output binding specificity, can be considered as reliable and potentially physically significant. Although high-affinity interactions can be nonspecific[38,53], interactions that contribute to high specificity would be expected to maximize binding affinity across different base pair possibilities. Therefore, the DeepPBS importance scores associated with these interactions should display some correlation with the corresponding binding affinities. We can test this hypothesis experimentally by using alanine scanning mutagenesis data (Supplementary Section 1). Sets

of such experimental data have been made available through recent contributions[54] in the field. Utilizing these data[42], we applied suitable filtering for our context and calculated the log sum aggregated residue level importance scores using DeepPBS (Methods).

A regression plot and Pearson's correlation coefficient (PCC), as shown in Fig. 4h, illustrate the correspondence between computed values and experimental ΔΔ$G$ values for a diverse array of proteins and residues within the protein–DNA interface (Supplementary Table 1). The obtained PCC of 0.60 corroborates our hypothesis. It is noteworthy that the model was not trained to predict these values. These values were only obtained through perturbing the wild-type (WT) structures as input (Supplementary Fig. 4 and Supplementary Table 1). These results highlight the potential of DeepPBS as an economical guide for experimentalists who are selecting alanine scanning mutagenesis experiments to conduct at the protein–DNA interface.

### Application to designed scaffolds targeting specific DNA

Recent work[34] made significant progress in designing structural models of fully synthetic helix-turn-helix (HTH) protein scaffolds targeting specific DNA sequences. We applied DeepPBS to synthetically designed proteins targeting a specific DNA sequence (GCAGATCTGCACATC), named DBP5/6/9/35, respectively (Fig. 5a,e,i,m). The predicted PWMs are shown (Fig. 5b,f,j,n) and the heavy atom level RI scores are visualized for the interfaces (Fig. 5c,g,k,o). We explored qualitative agreement of these predictions with experimental results obtained from the study (Fig. 5d,h,l,p, relative binding signal of all possible single base-pair mutations obtained via flow cytometry analysis[34] in yeast display competition assays). DeepPBS mostly correctly predicted the columns of high specificity (where the mutants show less binding that is darker red) except for a couple of cases. Some of the alternate base preference predictions by DeepPBS appear to agree with the experimental data. For example, for DBP35-position 11, DeepPBS predicts an alternate specific binding possibility to C along with the WT base A, and similarly for DBP35-position 9 and DBP5-position 7. Also, it is important to look at the flanking predictions for DeepPBS' ability to produce sensible predictions for unbound DNA regions. For DBP9 and DBP6, the flanking predictions look remarkably uniform, which is consistent with the designed structure having mostly unbound canonical B-DNA structure. This baseline behavior is intuitive and nontrivial in this problem setting (given that there is a DNA sequence present in the design and the model has to circumvent overfitting of it). On the other hand, for DBP5 and DBP35, the flanks have a non-canonical shape with a narrow minor groove interaction with a loop region of the protein (obtained from PDB ID 1L3L). The DeepPBS prediction of a mostly A-tract preference (positions 3–8) is consistent with narrow minor groove preferred by such sequences[55]. DNA shape prediction[56] for the top base prediction of these columns (AAATTT) is consistent with the shape visualized in the design (Supplementary Fig. 12), showing a significant dip in minor groove width. These examples illustrate the potential for DeepPBS as a computational guide to performing expensive and laborious wet lab experiments.

### Discussion

Computationally identifying which DNA sequences, a given protein will bind to remains a challenging question. Although proteins from certain DNA-binding families, such as homeodomain[17,22,57,58] and $C_2H_2$ zinc finger proteins[16–18,20,40,59], have been studied extensively in this regard, a generalized model of binding specificity remains elusive. This complexity emanates, in part, from the pivotal role that the protein and DNA conformation or shape play in the context of binding specificity. For example, TBX5 undergoes an α- to $3_{10}$-helix conformational change when interacting with DNA. Despite the energy penalty, this transformation, in conjunction with an appropriately matching DNA shape, instigates a strong phenylalanine-sugar ring stacking, thereby facilitating binding[4]. Another example is the Trp repressor protein,

which exhibits an almost entirely geometry-driven binding specificity. This protein only forms direct and water-mediated H-bonds with the backbone phosphates[60], and the DNA shape required for optimal binding gives rise to sequence specificity. Capturing such interactions and how they lead to binding specificity with protein information alone is complicated and cannot be understood in a sequence space alone[24,61]. Furthermore, for many protein families, the protein monomer is insufficient[49] for binding; a biological assembly, potentially with other interaction partners[62], is often necessary.

DeepPBS achieves generality across protein families with the tradeoff of requiring a docked sym-helix, representing a significant step toward solving the larger unsolved problem. As demonstrated in this work, coupling DeepPBS with attempts to model protein–DNA complexes provides a significant step forward in predicting binding specificity across families, based solely on protein information.

DeepPBS allows exploration of exciting future possibilities, including the creation of DNA-targeted protein designs that could potentially contribute to therapeutic advancements. DeepPBS could serve as a preliminary screening tool for devised candidate complexes, ensuring their specificity to the intended target DNA sequence before any costly experimental validations. Moreover, recent studies have shown that transcription factor–DNA binding can energetically favor mismatched base pairs[63]. Given the combinatorial complexity of possible hypotheses, deciding which DNA mismatch experiments to perform to discover more such instances poses a significant challenge. Although there is currently a lack of training data for base-pair mismatches, the DeepPBS architecture, in theory, could facilitate the prediction of mismatched base-pair binding specificity. This approach could assist in deciding which experiments to conduct.

In summary, we have introduced a computational framework that distills the intricate structural nuances of protein–DNA binding and bridges this understanding with binding specificity data, effectively connecting structure-determining and specificity-determining experiments. The DeepPBS architecture allows inspection of family-specific 'groove readout' and 'shape readout' patterns and their effects on binding specificity. Although structure prediction methods like RFNA[29], MELD-DNA[31] and AlphaFold3 (ref. 32) can predict a complex from given protein and DNA sequences, they cannot provide insights into binding specificity. The development of these computational methods for structure prediction expands the need of an approach like DeepPBS to derive protein–DNA binding specificity. DeepPBS operates on predicted complexes to yield the binding specificity of the system, thereby guiding the further improvement of modeling techniques for protein–DNA complexes. DeepPBS, despite its generality, exhibits performance comparable to the recently described family-specific method rCLAMPS[17]. In addition to modeled complexes for biologically existing systems, DeepPBS is also applicable to in silico synthetically designed proteins that target specific DNA sequences.

DeepPBS-derived RI scores are biologically relevant. They can be aggregated at a protein residue level, aligning with alanine scanning mutagenesis experimental data. Another advantage of DeepPBS is its speed in predicting binding specificity. Specifically, DeepPBS only requires a single forward call through the model (no required database search or multiple sequence alignment computation), making it suitable for high-throughput applications such as analyzing MD simulation trajectories (Supplementary Section 10 and Supplementary Fig. 6). In this context, DeepPBS is robust to small dynamical fluctuations and can respond to conformational changes (Supplementary Video 1).

The current version of DeepPBS has inherent limitations. It is tailored for double-stranded DNA and is not yet applicable to single-stranded DNA, RNA or chemically modified bases. However, there is potential for extending the model to accommodate these different scenarios as well as other polymer–polymer interactions and potentially for mechanistic mutations. Further limitations include data limitations, as discussed in Supplementary Section 12. The DeepPBS

architecture can be refined and expanded in terms of applications and engineering enhancements. Collectively, these possibilities hint at an exciting future for molecular interaction studies and computationally driven synthetic biology.

## Online content

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

## Methods

### Data sources

The dataset used for training was assembled by integrating protein–DNA structures from the PDB and their corresponding PWMs from JASPAR (2022)[36] and HOCOMOCO (V11)[37]. These two databases were selected for their accessibility, comprehensive collection and nonredundancy. The detailed description can be found in Supplementary Section 1 and Supplementary Fig. 1.

### Cross-validation regimen

A fivefold cross-validation set was constructed with 523 data points as described in Supplementary Section 1. Each datapoint corresponds to a biological assembly containing a protein chain with a corresponding PWM sampled from either JASPAR or HOCOMOCO. The PWM is aligned to DNA in the structure to create a correspondence for loss/metric calculation purposes using an ungapped local alignment process (Supplementary Section 2, 'Performance Metrics'). For each fold, cross-validation predictions were made by a model (same for other variations as in Supplementary Fig. 5a) trained on the remaining four folds (reported in Supplementary Fig. 5a–c). Full details of training can be found in Supplementary Section 6.

### Benchmark regimen

Datapoints not included in the cross-validation folds were resampled to create a separate benchmark dataset (biological assemblies corresponding to 130 protein chains). This sampling followed the same quality criterion described in Supplementary Section 1, and up to five members per cluster were sampled. Ensemble average predictions of models trained on cross-validation folds are reported for this set in Fig. 2a–d. Combined preprocessing and inference time for one biological assembly is on the order of seconds (for example, for PDB ID 5X6G, about 15–20 s). The DeepPBS ensemble described here was used for all applications of the predicted structures.

### PWM

For the purposes of this study, a PWM is defined as an $N \times 4$ matrix, where $N$ represents the length of the DNA of interest, and the four positions correspond to the four DNA bases: adenine (A), cytosine (C), guanine (G) and thymine (T). Each column in the PWM represents the probabilities of the four bases occurring at that particular position.

$$\text{Col}_{\text{PWM}} = [P_A, P_C, P_G, P_T]$$

$$P_A + P_C + P_G + P_T = 1$$

### DNA symmetrization

The DNA representation used is carefully designed with several considerations. First, the DNA sequence in the input complex might not correspond to a high-affinity sequence, particularly in designed structures. Second, an all-atom graph representation, similar to the protein, is not convenient because the model ultimately needs to predict a 1D representation (that is, the PWM) that describes binding specificity. Third, structural data are sparse, and the exact atomic conformation of a bound DNA sequence can make the model overly sensitive and less useful.

Considering these factors, we represent the DNA in a base-symmetrized manner. As shown in Fig. 1c and Supplementary Fig. 2, this is achieved by designing a symmetrization schema in the base-pair frame, which symmetrizes the seven key atomic interaction positions (four in the major groove and three in the minor groove)[24]. Additionally, four positions are assigned for the sugar and phosphate moieties. For full details of this process, see Supplementary Section 3.

### DeepPBS architecture and training details

Detailed description of the DeepPBS architecture can be found in Supplementary Section 5. Training, cross-validation and benchmarking details are available in Supplementary Section 6.

### Performance metrics

Performance metrics used in this work are MAE and RMSE, defined as

$$\text{MAE}(Y, Y^{\text{pred}}) = \frac{1}{N} \sum_{i \in \{0..N-1\}} \sum_{b \in [A,C,G,T]} \left| Y_{ib} - Y_{ib}^{\text{pred}} \right|$$

$$\text{RMSE}(Y, Y^{\text{pred}}) = \sqrt{\frac{1}{N} \sum_{i \in \{0..N-1\}} \sum_{b \in [A,C,G,T]} \left( Y_{ib} - Y_{ib}^{\text{pred}} \right)^2}.$$

$N$ refers to the number of columns in the PWMs being compared. Both metrics follow 'the lower the better' principle. They are not independent but have different properties. A further discussion of the metrics is presented in Supplementary Section 7 and Supplementary Fig. 10.

### Bipartite edge perturbation and protein heavy atom importance score calculation

Supplementary Fig. 4 schematically describes the bipartite edge perturbation process for calculating protein heavy atom (say, atom $a$) importance scores. Briefly, the prediction is calculated twice: once (say, $Y_a$) while considering edges corresponding to the protein heavy atoms, and again (say, $Y_{\sim a}$) while masking the same edges. This process results in differences in predictions, which can be calculated using the mean absolute difference measure. On their own, these values may not be meaningful, but they can be normalized to the 0–1 range by dividing by the maximum value within a structure. The normalized values, RI scores, signify how much the specificity prediction is influenced by interactions made by the corresponding heavy atom. Depending on the downstream use, RI scores can be aggregated at the residue level using either the average, max or sum aggregations. Mathematically,

$$\text{RI}_a = \frac{\text{MAE}(Y_a, Y_{\sim a})}{\max\limits_{\{b \in \text{all atoms}\}} \text{MAE}(Y_b, Y_{\sim b})}.$$

Computationally, this process is like measuring the effect of a deactivating mutation, which is why we hypothesized that, at a residue level, these scores could correlate with alanine scanning mutagenesis data. For comparison with alanine scanning mutagenesis experiments (Fig. 4h) at a residue level, the log sum aggregated importance score was calculated. For each atom $a$ of a residue $r$ in the protein–DNA interface, let the calculated RI be $\text{RI}_a$. Then, this value is calculated as

$$\text{LogSum aggregated residue importance}\,(r) = \log_2\left(1 + \sum_{a \in r} \text{RI}_a\right).$$

Structure visualizations presented were produced using PyMOL2.5.

### Description of competitor assay for quantifying designed proteins' binding specificity

Glasscock et al.[34] used a yeast display assay to quantify binding of their designed proteins. The proteins were expressed by integrating the corresponding synthetic oligonucleotide to a yeast surface expression vector. Yeast cells expressing designed proteins on their surface were labeled with biotinylated dsDNA targets, streptavidin–phycoerythrin and anti-c-Myc fluorescein isothiocyanate in a 96-well plate format, after which a binding signal was quantified on an Attune NxT flow cytometer. Excess addition of a competitor nonfluorescent target DNA reduces this binding signal. Thus, scanning single mutations for each position was possible through the competitor producing the data shown in Fig. 5d,h,l,p.

### DeepPBS webserver

DeepPBS is available as a webserver at https://deeppbs.usc.edu. The webserver provides the functionality of the DeepPBS method

of predicting a PWM on the basis of the structure of a protein–DNA complex. The structure can be uploaded as a PDB or macromolecular crystallographic information file. The webserver provides a documentation for users.

## Reporting summary

Further information on research design is available in the Nature Portfolio Reporting Summary linked to this article.

## Data availability

Datasets used for all analysis and associated custom scripts were deposited via figshare at https://doi.org/10.6084/m9.figshare.25678053 (ref. 64). Accession codes for discussed structures from the PDB: 1L3L, 7CLI, 2R5Z, 1CIT, 1F4K, 1GJI, 1TC3, 2BSQ, 2C9L, 5ZGN, 1BBX, 1KLN, 1N5Y, 5YUZ, 1QAI, 1XC8, 6T8H, 4TUI, 1DH3, 7OH9 and 1APL. UniProt accession codes for protein sequences discussed (folded with RFNA): Q8IUE0, Q6H878, O43680 and Q4H376. Accession codes for discussed experimental specificity data from JASPAR2022 and HOCOMOCOv11: MA1897.1, MA1568.1, MA1031.1, MA1572.1, MA0112.2, MA0112.3, ESR1_HUMAN.H11MO.0 and NFKB2_HUMAN.H11MO.0.B. Mutagenesis experiment data used are available from the SAMPDI website (http://compbio.clemson.edu/media/download/SAMPDI_dataset.xlsx). MELD-DNA modeled complex data were taken from Zenodo at https://doi.org/10.5281/zenodo.7501937 (ref. 65). Source data are provided with this paper.

## Code availability

Installable source code, pretrained models, associated guidelines and various custom scripts can be found via GitHub at https://github.com/timkartar/DeepPBS. The implementation is also available via a Code Ocean capsule at https://doi.org/10.24433/CO.0545023.v2. In addition, DeepPBS is accessible as a webserver through https://deeppbs.usc.edu.

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

## Acknowledgements

This work was supported by an Andrew J. Viterbi Fellowship in Computational Biology and Bioinformatics (to R.M.), a Washington Research Foundation postdoctoral fellowship (to C.J.G.), the Human Frontier Science Program (grant RGP0021/2018 to R.R.) and the National Institutes of Health (grant R35GM130376 to R.R.). We acknowledge L. Manna for setup and maintenance of the DeepPBS webserver and thank all Rohs lab members for support and valuable feedback.

## Author contributions

R.M., J.M.S. and R.R. conceived the project idea with input from T.P.C. R.M., J.M.S. and J.L. designed the model. R.M. and J.M.S. performed data preprocessing. R.M., with input from J.L. and J.M.S., performed model training and benchmarking. R.M., J.L. and T.P.C. developed all application ideas. R.M., J.L. and Y.J. carried out all applications and data analysis. A.S.C., with input from R.M., designed and built the web-based implementation. C.J.G. provided data for validation and application on predicted and synthetic designs. R.M., J.L., Y.J. and R.R. wrote the paper. All authors read and commented on the paper. R.R. supervised the project.

## Competing interests

The authors declare no competing interests.

## Additional information

**Correspondence and requests for materials** should be addressed to Remo Rohs.

# Reporting Summary

## Statistics

For all statistical analyses, confirm that the following items are present in the figure legend, table legend, main text, or Methods section.

| n/a | Confirmed | |
|---|---|---|
| ☐ | ☒ | The exact sample size (*n*) for each experimental group/condition, given as a discrete number and unit of measurement |
| ☒ | ☐ | A statement on whether measurements were taken from distinct samples or whether the same sample was measured repeatedly |
| ☐ | ☒ | The statistical test(s) used AND whether they are one- or two-sided<br>*Only common tests should be described solely by name; describe more complex techniques in the Methods section.* |
| ☒ | ☐ | A description of all covariates tested |
| ☒ | ☐ | A description of any assumptions or corrections, such as tests of normality and adjustment for multiple comparisons |
| ☐ | ☒ | A full description of the statistical parameters including central tendency (e.g. means) or other basic estimates (e.g. regression coefficient) AND variation (e.g. standard deviation) or associated estimates of uncertainty (e.g. confidence intervals) |
| ☒ | ☐ | For null hypothesis testing, the test statistic (e.g. *F*, *t*, *r*) with confidence intervals, effect sizes, degrees of freedom and *P* value noted<br>*Give P values as exact values whenever suitable.* |
| ☒ | ☐ | For Bayesian analysis, information on the choice of priors and Markov chain Monte Carlo settings |
| ☒ | ☐ | For hierarchical and complex designs, identification of the appropriate level for tests and full reporting of outcomes |
| ☐ | ☒ | Estimates of effect sizes (e.g. Cohen's *d*, Pearson's *r*), indicating how they were calculated |

*Our web collection on statistics for biologists contains articles on many of the points above.*

## Software and code

Policy information about availability of computer code

| Data collection | Data was downloaded from publicly available datasets. (PDB, UniProtKB, JASPAR2022, HOCOMOCOv11, SAMPDI, MELD-DNA) |
|---|---|
| Data analysis | All core DeepPBS (https://github.com/timkartar/DeepPBS) code was written in python3.9+ with various pythonic dependencies (full list available at https://github.com/timkartar/DeepPBS/blob/main/deeppbs_linux.yml). Packages used for geometric deep learning are pytorch1.12+ and torch-geometric(pyg v2.0+). Data analysis and visualization was carried out in same environment. Structure visualizations presented in the manuscript are done using PyMOL2.5. 3DNAv2.3 and Curves5 was used in pre-processing steps (executables are provided https://github.com/timkartar/DeepPBS/tree/main/dependencies/bin). CD-HITv4.8.1 was used for protein sequence clustering. RFNA v0.2 was used for complex structure prediction. MD simulation was performed using Gromacs2020.3, LINCS algorithm included with Gromacs2020.3 was employed to constrain all bonds. HMMERv3.4 was used for homeobox detection. https://github.com/jlwetzel-slab/rCLAMPS and corresponding instructions were used for predictions by rCLAMPS (github commit version 32a94edb65e87c6d038823dc34c4bcf6e1071b7b ) method. Installable source code, pre-trained models, associated guidelines and various custom scripts can be found at https://github.com/timkartar/DeepPBS. The implementation is also available via a Code Ocean capsule at https://doi.org/10.24433/CO.0545023.v2. In addition, DeepPBS is accessible as a web server through https://deeppbs.usc.edu. |

For manuscripts utilizing custom algorithms or software that are central to the research but not yet described in published literature, software must be made available to editors and reviewers. We strongly encourage code deposition in a community repository (e.g. GitHub). See the Nature Portfolio guidelines for submitting code & software for further information.

## Data

Policy information about availability of data

All manuscripts must include a data availability statement. This statement should provide the following information, where applicable:
- Accession codes, unique identifiers, or web links for publicly available datasets
- A description of any restrictions on data availability
- For clinical datasets or third party data, please ensure that the statement adheres to our policy

Datasets used for all analysis and associated custom scripts are deposited at https://doi.org/10.6084/m9.figshare.25678053 . Source data for figures and additional supplementary data are available. Accession codes for discussed structures from the Protein Data Bank: 2R5Y, 3Q05, 5X6G, 1L3L, 7CLI, 2R5Z, 1CIT, 1F4K, 1GJI, 1TC3, 2BSQ, 2C9L, 5ZGN, 1BBX, 1KLN, 1N5Y, 5YUZ, 1QAI, 1XC8, 6T8H, 4TUI, 1DH3, 7OH9, 1APL. UniProt accession codes for protein sequences discussed (folded with RFNA) Q8IUE0, Q6H878, O43680, Q4H376 . Accession codes for discussed experimental specificity data from JASPAR2022 and HOCOMOCOv11: MA1897.1, MA1568.1, MA1031.1, MA1572.1, MA0112.2, MA0112.3, ESR1_HUMAN.H11MO.0, NFKB2_HUMAN.H11MO.0.B. Mutagenesis experiment data used is available from SAMPDI website. MELD-DNA modeled complex data was taken from doi: 10.5281/zenodo.7501937.

## Human research participants

Policy information about studies involving human research participants and Sex and Gender in Research.

| | |
|---|---|
| Reporting on sex and gender | N/A |
| Population characteristics | N/A |
| Recruitment | N/A |
| Ethics oversight | N/A |

Note that full information on the approval of the study protocol must also be provided in the manuscript.

# Field-specific reporting

Please select the one below that is the best fit for your research. If you are not sure, read the appropriate sections before making your selection.

☒ Life sciences          ☐ Behavioural & social sciences          ☐ Ecological, evolutionary & environmental sciences

For a reference copy of the document with all sections, see nature.com/documents/nr-reporting-summary-flat.pdf

# Life sciences study design

All studies must disclose on these points even when the disclosure is negative.

| | |
|---|---|
| Sample size | We attempted to gather the largest, non-redundant dataset possible. All sample sizes for various datasets used in the study are provided through the manuscript and supplementary information. |
| Data exclusions | All filtering criteria were predetermined. Sample sizes were constrained by processing abilities of the various softwares used in the study. All descriptions for the same are provided through the manuscript and supplementary information. |
| Replication | Repeated model training and analysis were carried out at various intervals throughout the design of the study, fine tuning and through different stages of revision, leading to same conclusions. |
| Randomization | Sampling was involved in creation of cross validation dataset. For each sequence cluster, biological assemblies corresponding to upto 5 members were randomly sampled into one of five cross validation folds. Each model was trained on four folds and validated on the fifth fold. Biological assemblies not sampled into cross validation folds were resampled into a benchmark set. |
| Blinding | Standard five fold cross validation technique was used to train the models and a separate fully blindfold benchmark set was kept aside. |

# Reporting for specific materials, systems and methods

We require information from authors about some types of materials, experimental systems and methods used in many studies. Here, indicate whether each material, system or method listed is relevant to your study. If you are not sure if a list item applies to your research, read the appropriate section before selecting a response.

## Materials & experimental systems

| n/a | Involved in the study |
|-----|----------------------|
| ☒ | Antibodies |
| ☒ | Eukaryotic cell lines |
| ☒ | Palaeontology and archaeology |
| ☒ | Animals and other organisms |
| ☒ | Clinical data |
| ☒ | Dual use research of concern |

## Methods

| n/a | Involved in the study |
|-----|----------------------|
| ☒ | ChIP-seq |
| ☒ | Flow cytometry |
| ☒ | MRI-based neuroimaging |

