## [Peer Review File · Nature Methods]

Peer Review Information

Manuscript Title: Geometric deep learning of protein-DNA binding specificity

Corresponding author name(s): Remo Rohs

Editorial Notes: None

Reviewer Comments & Decisions:

Decision Letter, initial version:

Dear Remo,

Your Article, "Geometric deep learning for interpretable prediction of protein–DNA binding specificity", has now been seen by 3 reviewers. As you will see from their comments below, although the reviewers find your work of considerable potential interest, they have raised some concerns. We are interested in the possibility of publishing your paper in Nature Methods, but would like to consider your response to these concerns before we reach a final decision on publication.

We therefore invite you to revise your manuscript to address these concerns. In addition to addressing all the technical and text related comments, we think a proper review of the software/code related to your work will be important. We recommend providing the code in a more accessible manner for both testing and for the users. We recommend the use of the Code Ocean platform for peer review of code and will be happy to send you more information if you are interested.

- * include a point-by-point response to the reviewers and to any editorial suggestions
- * please underline/highlight any additions to the text or areas with other significant changes to facilitate review of the revised manuscript
- * address the points listed described below to conform to our open science requirements
- * ensure it complies with our general format requirements as set out in our guide to authors at

www.nature.com/naturemethods

* resubmit all the necessary files electronically by using the link below to access your home page

[Redacted]

We hope to receive your revised paper within 6 weeks. If you cannot send it within this time, please let us know. In this event, we will still be happy to reconsider your paper at a later date so long as nothing similar has been accepted for publication at Nature Methods or published elsewhere.

OPEN SCIENCE REQUIREMENTS

REPORTING SUMMARY AND EDITORIAL POLICY CHECKLISTS

DATA AVAILABILITY

All novel DNA and RNA sequencing data, protein sequences, genetic polymorphisms, linked genotype and phenotype data, gene expression data, macromolecular structures, and proteomics data must be deposited in a publicly accessible database, and accession codes and associated hyperlinks must be provided in the "Data Availability" section.

CODE AVAILABILITY

Please include a "Code Availability" subsection in the Online Methods which details how your custom code is made available. Only in rare cases (where code is not central to the main conclusions of the paper) is the statement "available upon request" allowed (and reasons should be specified).

For more information on our code sharing policy and requirements, please see: <https://www.nature.com/nature-research/editorial-policies/reporting-standards#availability-of-computer-code>

MATERIALS AVAILABILITY

ORCID

Nature Methods is committed to improving transparency in authorship. As part of our efforts in this direction, we are now requesting that all authors identified as 'corresponding author' on published papers create and link their Open Researcher and Contributor Identifier (ORCID) with their account on the Manuscript Tracking System (MTS), prior to acceptance. This applies to primary research papers only. ORCID helps the scientific community achieve unambiguous attribution of all scholarly contributions. You can create and link your ORCID from the home page of the MTS by clicking on 'Modify my Springer Nature account'. For more information please visit please visit www.springernature.com/orcid.

Sincerely,
Arunima

Arunima Singh, Ph.D.
Senior Editor
Nature Methods

Reviewers' Comments:

Reviewer #1:

Remarks to the Author:

This article presents a deep learning approach that takes a protein-DNA co-complex structure and predicts the protein's DNA-binding specificity as a position weight matrix. This is a central problem in gene regulation and protein interaction specificity and is of broad interest, and the authors present an original and interesting approach, along with a range of downstream applications.

There are some things that need to be addressed before publication:

(1) Since this is a machine learning approach, sequence similarity of the proteins has to be considered in the cross validation (as similar proteins have similar DNA-binding specificities). Highly sequence similar DNA binding domains should be in the same fold (ideally dealing with identical or near-identical

base contacting positions)

(2) The paper should show how much performance drops on the benchmark dataset if modeled structures are used, as this would be the most common use case of the method - and the results should be in the main paper.

(3) While MAE is useful for comparing variations of the method and other related approaches, I don't find it particularly helpful in determining whether good predictions of DNA-binding specificity are made. For example (1, 0, 0, 0) and (0.5, 0.5, 0, 0) has -MAE of -1, but so does (.25, .25, .25, .25) and (.5, .5, 0, 0) — about the median performance given in Fig 3D for bZIPs, but I would consider the first case OK agreement and the second case bad agreement. The authors should either give examples in the supplement of PWM pairs giving a range of MAE scores, as well an acceptable threshold to declare success, or use something like Pearson or Spearman correlations which are easier to interpret

Reviewer #3:

Remarks to the Author:

Predictions of protein-DNA binding specificity based on protein (or protein-DNA) structures are challenging. Some family-specific methods exist, for structural families with abundant structures, but they cannot be generalized beyond the families they were trained on. General methods would be useful especially in the context of predicting the effects of genetic variants, where it is highly preferable to not have to generate new structures experimentally for every variant of interest. The method described here, DeepPBS, is an important step in this direction.

DeepPBS uses an interesting representation of 3D protein-DNA structures as a bipartite graph, with heavy atoms as vertices, and using distinct representation for the protein and the DNA. The design is flexible in that it can include just "groove readout" (i.e. specificity coming mostly from sequence), "shape readout" (i.e. specificity coming from DNA shape), or both. In addition, it allows the explicit incorporation of sequence information, which can further improve performance. An important feature of DeepPBS is that the resulting models can be used to extract interpretable information about which residues are interacting with the DNA (Figure 4).

The authors evaluate DeepPBS mostly using a cross-validation approach. In addition, the manuscript includes a good description of how DeepPBS compares to existing methods such as rCLAMPS; briefly, the performance is similar (Figure 3h), but DeepPBS has the advantage that it can predict more than just one protein family, which is a limitation of current methods. Finally, it is important to note that DeepPBS can be used only with real structures, but also with predicted structures from recent methods such as AlphaFold and RoseTTAFold.

Overall, DeepPBS represents an important contribution that leverages state-of-the-art machine learning approaches combined with a clever representation of 3D DNA and protein structures to predict protein-DNA binding specificity using structural data across protein families.

Minor comments:

1) The paper is quite dense, but at the same time well written. Still, some additional details should be included in the main text, especially in regards to the validation. For example, in Figure 2a, it is

completely unclear what the shaded region is, without reading the supplementary material. The information presented in the figure should be clear from the main text alone.

2) In the same figure (2a), the authors should include information on whether the performance of different DeepPBS models is significantly different (e.g. between "DeepPBS groove readout" and "DeepPBS"). Claims are made in the text about this, without statistical support.

3) Also in Figure 2, outliers are not shown in panels a and d. While the message of the manuscript would not change with the addition of the outliers, they should be shown (maybe in a supplementary figure) and a brief discussion about the outliers would be helpful for the readers to understand the drawbacks of DeepPBS.

4) Finally, why is MAE used in some plots, and RMSE in others? And why is MAE preferred? This should be clarified.

Reviewer #4:

Remarks to the Author:

This manuscript introduces a computational pipeline, DeepPBS, which incorporates structural features into the inference and prediction of target DNA sequence selectivity by nucleoprotein complexes. In particular, DNA geometric information was encoded by a coarse-graining procedure termed "sym-helix" that removes base-specific chemical details. Various experimentally and computationally derived datasets were used to illustrate the pipeline's capability to discern and predict biophysical drivers of sequence specificity. A "relative importance" (RI) metric is devised to quantify the comparative contribution of individual residues to DNA specificity. Compared to the closest competitor, rCLAMPS, a major advantage of DeepPBS appears to be that it is both structure-aware but also agnostic to specific structural families.

The main text is primarily descriptive. Much more informative is the Supplementary material. Based on the exposition there, the parameterization of various structural features as well as training and validation of the model seem rigorous, and the SI figures S7 and S8 could be particularly useful for non-specialist readers to gauge the usefulness of the pipeline.

In approximate order of importance, questions to the authors are as follows:

1. From reading the text, it is not clear how structural data from wildtype and mutant proteins are identified and handled. The SI discussed alanine scanning mutagenesis, but this is a very specific type of mutation. Mechanistic mutations e.g., charge reversal, Lys-to-Leu are myriad in diversity and scale (from single residues to entire secondary structures), and inclusion in numbers may bias model training.

2. In head-to-head comparisons with experimental ensembles (e.g., HOCOMOCO for NF- κ B2 in Figure S5), it would help if the distributions of bases over positions could be statistically quantified and compared. By inspection, the DeepPBS prediction is more "nuanced" than HOCOMOCO (and certainly the crystallographically derived set) but it is not obvious how the results should be comparatively summarized.

3. Acronyms for metrics (e.g., MAE) and their definitions are given either near the end of the main text (in Methods) or in the SI, which is out of order for the naïve reader.

4. Related to Figure 2: The nomenclature of "DeepPBS" (alone) as the combination of "DeepPBS Shape Readout" and "DeepPBS Groove Readout" could be improved to be more obvious (Fig 2a; Line 26, p.5).

5. Figure S7h: What are the red spheres?

COMMENT ON THE CODE: For this review, difficulties were encountered in executing the codes. For review purposes, attempts were made to install DeepPBS on a fresh linux (Ubuntu 22.04) installation, following the instructions provided on the provided github link. Installation of Anaconda and various dependencies went fine but executing the DeepPBS code resulted in crashes related to the PyTorch dependencies, ending either in non-recognition of the GPU (under CUDA 12.2), or inability to proceed in a CPU-only setup. This was not resolved by tinkering version numbers with conda updates and in the environmental (.yml) file. Likely conflicts between evolving versions of PyTorch and CUDA are responsible, and should be addressable by the authors. Nevertheless, for the purpose of the present review, the code did not run as instructed. I might suggest that the authors incorporate some provisions in the package (or at least explicit instructions) to help future-proof the code for users who could not be expected to have access to legacy versions of conda and the various dependencies.

Author Rebuttal to Initial comments

RESPONSE TO REVIEWERS

Manuscript NMETH-A52876A

Reviewers' Comments:

Reviewer #1:

Remarks to the Author:

This article presents a deep learning approach that takes a protein-DNA co-complex structure and predicts the protein's DNA-binding specificity as a position weight matrix. This is a central problem in gene regulation and protein interaction specificity and is of broad interest, and the authors present an original and interesting approach, along with a range of downstream applications.

AUTHORS: We thank the Reviewer for their assessment of our work as original and interesting.

There are some things that need to be addressed before publication:

(1) Since this is a machine learning approach, sequence similarity of the proteins has to be considered in the cross validation (as similar proteins have similar DNA-binding specificities). Highly sequence similar DNA binding domains should be in the same fold (ideally dealing with identical or near-identical base contacting positions)

AUTHORS: We thank the Reviewer for raising this important point. We are happy to report that this is already considered. We do recognize that this point was not well emphasized in the dataset description and we have updated the manuscript with a new section, **Supplementary Section 11: Measures ensuring prevention of overfitting to protein sequences**, to reflect this. We also have now provided a full list of the protein-to-fold assignment through the **Extended Data file (.xlsx)**, and the sampling code is available via GitHub as specified in the manuscript's **Code Availability** section.

Inspired by this important comment, we also performed a new experiment, where we trained a model after fully shuffling the cross-validation set. The shuffled folds increased the performances (Table R1) compared to what we report in the manuscript using our cluster-based separated cross-validation folds (following the Reviewer's recommendation).

	Cross-validation Metrics	MAE (lower is better)	Avg PCC (higher is better)	RMSE (lower is better)	Avg Information Content Difference (lower is better)
DeepPBS	Clusterwise folds	0.68 ± 0.25	0.56 ± 0.23	0.53 ± 0.17	0.56 ± 0.19
	Shuffled folds	0.64 ± 0.24	0.59 ± 0.23	0.51 ± 0.17	0.53 ± 0.18
DeepPBS with DNA SeqInfo	Clusterwise folds	0.44 ± 0.15	0.808 ± 0.13	0.37 ± 0.13	0.47 ± 0.14
	Shuffled folds	0.42 ± 0.16	0.812 ± 0.13	0.36 ± 0.13	0.45 ± 0.15

Table R1:

In addition, we would like to emphasize other measures that we implemented in our model to prevent overfitting on protein sequences: The input of DeepPBS is purely structural and physico-chemical, and does not contain a direct sequence representation. On the other hand, only up to five members were chosen per cluster for a fold. These structures may demonstrate different spatial conformations interacting with potentially different DNA sequence or shape and are combined with randomly picked target PWM from JASPAR or HOCOMOCO, whenever possible. These guardrails in architecture and all of these variations, even for similar protein sequences, makes the model less prone to overfitting on protein sequences given no explicit sequence information.

(2) The paper should show how much performance drops on the benchmark dataset if modeled structures are used, as this would be the most common use case of the method - and the results should be in the main paper.

AUTHORS: We are very thankful to the Reviewer for pointing out this important experiment. As advised, we now used RoseTTAFoldNA to fold the benchmark dataset. 108 out of 130 modeled structures were folded to structures processable by our pre-processing pipeline (the remaining structural models from RoseTTAFoldNA failed either helix detection and/or shape parameter and other feature calculations). Among the 108 structures that resulted in structures that can be analyzed, 98 structures have the protein reasonably placed in relation to the DNA double helix, determined by atom-to-atom contact count (count > 500 with a 5 Å cutoff) . We compared the performance on this RFNA-folded set with the corresponding performance on the experimentally derived set. Additionally, we also included the performance for a high confidence subset of these RFNA-folded structures (pLDDT > 0.9, 31 structures). As recommended by the Reviewer, these results are now part of the main text as **Fig. 3g**. As expected, there is a drop in performance. However, we anticipate that protein-DNA complex modeling will improve in the future, with new unreleased models like AlphaFold-latest which claim to outperform RFNA.

In addition, we provided further case studies of applicability of DeepPBS on a fully synthetic DNA binding proteins, designed and provided to us by Glasscock et al. 2023 (PMID: **37790440**) which target a specific DNA. We now include DeepPBS predictions on these designed synthetic structures along with yeast display competition assay data, demonstrating the consistency between our prediction and experimental results. We discuss both success and non-success cases seen in these examples, along with the model's predictive behavior considering DNA shape in flanking regions present in the design. We think that these results demonstrate the potential of this model for guiding molecular design in synthetic biology. These results are now included in the main text too as a new **Fig. 5** (and **Fig. S12**).

Previous Fig. 5 on the DeepPBS application on MD and associated description were moved to Supplementary Information and combined with supplementary **Fig. S6** to shorten the text and reduce figure content in the main manuscript.

(3) While MAE is useful for comparing variations of the method and other related approaches, I don't find it particularly helpful in determining whether good predictions of DNA-binding specificity are made. For example (1, 0, 0, 0) and (0.5, 0.5, 0, 0) has -MAE of -1, but so does (.25, .25, .25, .25) and (.5, .5, 0, 0) — about the median performance given in Fig 3D for bZIPs, but I would consider the first case OK agreement and the second case bad agreement. The authors should either give examples in the supplement of PWM pairs giving a range of MAE scores, as well an acceptable threshold to declare success, or use something like Pearson or Spearman correlations which are easier to interpret.

AUTHORS: This is an excellent point. As the Reviewer suggested, we added a supplementary **Fig. S10a** demonstrating how the MAE metric behaves for various different target PWMs and

predictions. Based on these plots and the inherent performance limit computed for this metric (**Fig. 2a**), we can consider values less than 0.8 to be of reasonable agreement and below 0.6 to be in good agreement. We note that these values should not be set in stone, as the problem in question is a regression problem as opposed to binary classification.

Some further thoughts on this important topic as we actually considered this question thoroughly during this project: We agree with the Reviewer that the uniform prediction [0.25, 0.25, 0.25, 0.25] can be regarded as a ‘bad’ prediction, because a naïve model can always predict such a value. This prediction will perform well for highly non-specific binders (e.g., cases like in **Fig. S8**) but will fail for highly specific binders. However, the one-hot prediction [1,0,0,0] can also be a naïve (‘bad’) prediction for this problem setting. It is the case when the sequence present in the co-crystal structure itself is the output of the prediction, e.g., for the sequence ACG: [[1,0,0,0],[0,1,0,0],[0,0,1,0]]. For experimentally determined structures, this prediction will generally perform well, especially for highly specific binders, but will it fail for nonspecific binders. Ultimately, we wanted to create a general model that can handle specific binders and non-specific binders, so we needed a target metric to strike a balance between both scenarios.

Therefore, in our opinion, looking at the predictive performances in context of the alignment scores of the co-crystal structure derived sequence with the target PWM gives a clearer picture. This has now been added (for the benchmark set) to the manuscript as supplementary **Fig. S10b** (reproduced below).

We chose a continuous distance metrics like MAE over statistical measures like PCC (Pearson R) or SCC (Spearman R) because the statistical measures are known to be less robust for small sample sizes (<https://doi.org/10.2307/2279076>, <https://doi.org/10.1007/BF02294183>), in this case, resulting in four linearly dependent points. In our observation, SCC can only take a few discrete values in this scenario and can sharply change for a very small change of values that change the rank order. The PCC metric, although it assumes continuous values, is affected by similar non-intuitive situations. We can consider the following three slightly altered predictions as an example:

$\text{PCC}([0.5, 0.5, 0, 0], [0.25, 0.25, 0.25, 0.25]) = \text{Undefined}$

$\text{PCC}([0.5, 0.5, 0, 0], [0.23, 0.24, 0.26, 0.27]) = -0.9487$

$\text{PCC}([0.5, 0.5, 0, 0], [0.27, 0.26, 0.24, 0.23]) = 0.9487$

(calculated using *scipy.stats.pearsonr()* function)

Intuitively, all three predictions are of similar caliber. However, the PCC metric paints a dramatically different picture in each case. MAE on the other hand produces values of 1, 1.06 and 0.94. We can also easily construct other non-intuitive situations for PCC. For example, if the target is [0.1, 0.2, 0.3, 0.4], any monotonically increasing prediction will have a high PCC value.

For example, $\text{PCC}([0.1, 0.2, 0.3, 0.4], [0.001, 0.002, 0.003, 0.994]) = 0.776$

The PCC value of 0.776 gives a good impression about the prediction, while in reality, it is almost just a one-hot prediction. MAE, on the other hand, produces a value of 1.187 indicating a 'bad' prediction.

Additionally, for training a deep learning model we need a continuous and differentiable loss space. Thus, we prioritize the MAE metric because it helps us guide architectural choices of our model.

In our observation, RoseTTAFoldNA has a compromised folding performance for full length bZIP family proteins and we only claim that the first round predictions are better than the initial guess sequence.

We thank the Reviewer again for bringing up this excellent topic and we recognize the prior lack of discussion on this issue in the manuscript. We have, therefore, now included this discussion as **Supplementary Section 7**.

Reviewer #3:

Remarks to the Author:

Predictions of protein-DNA binding specificity based on protein (or protein-DNA) structures are challenging. Some family-specific methods exist, for structural families with abundant structures, but they cannot be generalized beyond the families they were trained on. General methods would be useful especially in the context of predicting the effects of genetic variants, where it is highly preferable to not have to generate new structures experimentally for every variant of interest. The method described here, DeepPBS, is an important step in this direction.

AUTHORS: We thank the Reviewer for their assessment of DeepPBS as an important step.

DeepPBS uses an interesting representation of 3D protein-DNA structures as a bipartite graph, with heavy atoms as vertices, and using distinct representations for the protein and the DNA. The design is flexible in that it can include just “groove readout” (i.e. specificity coming mostly from sequence), “shape readout” (i.e. specificity coming from DNA shape), or both. In addition, it allows the explicit incorporation of sequence information, which can further improve performance. An important features of DeepPBS is that the resulting models can be used to extract interpretable information about which residues are interacting with the DNA (Figure 4).

The authors evaluate DeepPBS mostly using a cross-validation approach. In addition, the manuscript includes a good description of how DeepPBS compares to existing methods such as rCLAMPS; briefly, the performance is similar (Figure 3h), but DeepPBS has the advantage that it can predict more than just one protein family, which is a limitation of current methods. Finally, it is important to note that DeepPBS can be used only with real structures, but also with predicted structures from recent methods such as AlphaFold and RoseTTAFold.

AUTHORS: We thank the Reviewer for their succinct summary of the advances of DeepPBS.

Overall, DeepPBS represents an important contribution that leverages state-of-the-art machine learning approaches combined with a clever representation of 3D DNA and protein structures to predict protein-DNA binding specificity using structural data across protein families.

AUTHORS: We are grateful for this view that DeepPBS represents an important contribution.

Minor comments:

1) The paper is quite dense, but at the same time well written. Still, some additional details should be included in the main text, especially in regards to the validation. For example, in Figure 2a, it is completely unclear what the shaded region is, without reading the supplementary material. The information presented in the figure should be clear from the main text alone.

RESPONSE: We considered the Reviewer’s comment carefully and decide to make the manuscript less dense by moving all methods and results that relate to the application of DeepPBS to molecular dynamics simulations to **Supplementary Information**, because this content was secondary to the manuscript. We thank the Reviewer also for pointing out the confusion regarding the shaded region in Fig. 2a. We have now updated the corresponding section (**Performance of DeepPBS on experimentally determined structures from PDB**) in the main text with the necessary details define these regions.

2) In the same figure (2a), the authors should include information on whether the performance of different DeepPBS models is significantly different (e.g. between “DeepPBS groove readout” and “DeepPBS”). Claims are made in the text about this, without statistical support.

AUTHORS: We thank the Reviewer for pointing this out. We recognize the lack of statistical measures supporting the figure. Through the Extended Data file we now have provided this information. In general, we would not expect the ‘DeepPBS Groove Readout’ and ‘DeepPBS’ models to be constituting a pair of statistically significantly different distributions (since Groove Readout is a rather strong determinant of binding specificity). However, this may be the case for just specific protein families. We have clarified the main text to make sure our claims are properly discussed. In addition, through an Extended Data spreadsheet, we have added statistical separation values for the plots in Fig. 2d too to paint a clearer picture.

We apologize if the text was misleading before, we have updated the text to reflect this (in Section: **Performance of DeepPBS on experimentally determined structures from PDB**).

3) Also in Figure 2, outliers are not shown in panels a and d. While the message of the manuscript would not change with the addition of the outliers, they should be shown (maybe in a supplementary figure) and a brief discussion about the outliers would be helpful for the readers to understand the drawbacks of DeepPBS.

AUTHORS: We have added the outliers for these results now. Thanks for suggesting that we explore these outliers. We specifically investigated and discussed the two outliers seen in Fig. 2a.

The outlier for the DeepPBS (Groove Readout) model is a TATA-box binding protein (TBP, only TBP occurring in this set) bound to nucleosome bound-DNA (PDB ID: 7OH9). TBP-DNA binding is known to be a primarily Shape Readout driven process depending on the strong bendability of the TATA motif (PMID: 16500964). The Groove Readout model is understandably unable to perform well in this case.

The outlier for the DeepPBS (both Groove Readout and Shape Readout) model is a homeodomain factor (PDB ID: **5Z2T**, MAE: 1.06). It is not an extreme outlier and reasonably close to the quartile mark. At first glance, it is unclear why this is an outlier. However, we realized there is another structure of the same protein in this set (PDB ID: **6A8R**, biological assembly 2) and DeepPBS performs, comparatively, much better for this one (MAE: 0.75). Upon investigation we realized the 5Z2T structure has a lower resolution and high degree of sidechain outliers (17.4%), according to the quality metrics provided by the wwPDB validation report. Therefore, we are not confident to make mechanistic inferences based on this structure.

Other than data quality limitations, another form of data limitation can be representation. For example, carboxylic acid side chains (glutamic and aspartic acids) are generally rare in biological DNA binding domains (due to electrostatic repulsion) and hence in DeepPBS training data. A synthetic biologist should be weary of this fact when designing domains with these residues.

We thank the Reviewer for this valuable suggestion and we have updated the manuscript with this discussion (Section in the main text: **Performance of DeepPBS on experimentally determined structures from PDB** section and **Supplementary Section 12**). We also expanded the text about limitations presented in the Discussion pointing to this section.

4) Finally, why is MAE used in some plots, and RMSE in others? And why is MAE preferred? This should be clarified.

AUTHORS: We primarily show MAE because it is a linear measure, corresponds to the L1 norm and the model was trained to optimize the L1 loss space. The only figure we showed with RMSE is Fig. 2b, because it illustrates the extent of bias towards the co-crystal structure derived sequence and PWM alignment score for the 'DeepPBS with DNA SeqInfo' model. Although the message we present does not change, the RMSE metric in this case shows the crossing point more clearly, since the biased model could be overfit in the L1 space. See below the two plots side by side for comparison: We have added the alternative version now in supplementary **Fig. S11**.

Reviewer #4:

Remarks to the Author:

This manuscript introduces a computational pipeline, DeepPBS, which incorporates structural features into the inference and prediction of target DNA sequence selectivity by nucleoprotein complexes. In particular, DNA geometric information was encoded by a coarse-graining procedure termed "sym-helix" that removes base-specific chemical details. Various experimentally and computationally derived datasets were used to illustrate the pipeline's capability to discern and predict biophysical drivers of sequence specificity. A "relative importance" (RI) metric is devised to quantify the comparative contribution of individual residues to DNA specificity. Compared to the closest competitor, rCLAMPS, a major advantage of DeepPBS appears to be that it is both structure-aware but also agnostic to specific structural families.

AUTHORS: We thank the Reviewer for their description of advances of DeepPBS.

The main text is primarily descriptive. Much more informative is the Supplementary material. Based on the exposition there, the parameterization of various structural features as well as training and validation of the model seem rigorous, and the SI figures S7 and S8 could be particularly useful for non-specialist readers to gauge the usefulness of the pipeline.

AUTHORS: We are thankful for highlighting the importance of the supplementary material for this manuscript. We tried our best to keep the manuscript readable for the general audience while providing additional details of our work in Supplementary Information.

In approximate order of importance, questions to the authors are as follows:

1. From reading the text, it is not clear how structural data from wildtype and mutant proteins are identified and handled. The SI discussed alanine scanning mutagenesis, but this is a very specific type of mutation. Mechanistic mutations e.g., charge reversal, Lys-to-Leu are myriad in diversity and scale (from single residues to entire secondary structures), and inclusion in numbers may bias model training.

AUTHORS: We thank the Reviewer for pointing this out. We apologize for the confusion. We would like to clarify that the model was not trained on any mutagenesis data. We did not handle any structural data for mutant proteins. We only used the wild type structural data as input and calculated relative importances of the atoms via edge perturbation (supplementary Fig. S4, Table S1). These values were then LogSum aggregated at a residue level. This is similar to measuring the effect of a deactivating mutation, but in silico. We hypothesized that these scores could correlate with alanine scanning mutagenesis data and found out they do correlate. We

agree with the Reviewer that without an extensive structural dataset of mechanistic mutations (containing corresponding co-crystal structure-PWM pairings), a venture to train a model to learn such effects will not be very successful.

This point has been clarified now in the text (Section: **Comparison of DeepPBS-derived residue-level importance with experimental data**), **Methods** section (under "*Bipartite edge perturbation and protein heavy atom importance score calculation*" and **Discussion** section).

2. In head-to-head comparisons with experimental ensembles (e.g., HOCOMOCO for NF- κ B2 in Figure S5), it would help if the distributions of bases over positions could be statistically quantified and compared. By inspection, the DeepPBS prediction is more "nuanced" than HOCOMOCO (and certainly the crystallographically derived set) but it is not obvious how the results should be comparatively summarized.

AUTHORS: We agree with the Reviewer that Fig. S5 was not properly quantified and labeled, and we apologize for the oversight. We have now added the standard information content view of the corresponding PWMs to help the reader. We also provided the pairwise MAEs between the sets for the reader to compare and have a more quantitative feeling of the result.

3. Acronyms for metrics (e.g., MAE) and their definitions are given either near the end of the main text (in Methods) or in the SI, which is out of order for the naïve reader.

AUTHORS: Thanks for pointing this out, we have now updated the Figure 2 caption with this information to make it easily available for the reader.

4. Related to Figure 2: The nomenclature of "DeepPBS" (alone) as the combination of "DeepPBS Shape Readout" and "DeepPBS Groove Readout" could be improved to be more obvious (Fig 2a; Line 26, p.5).

AUTHORS: We wish to keep the name of the 'DeepPBS' model the same as it is the model used to perform further applications presented in the manuscript. But we do recognize the point the Reviewer is making. We have updated Fig. 2a and by adding '(both Groove Readout and Shape Readout)' to make it clear that the 'DeepPBS' model includes both readout modes. We have made the same clarification in the text referenced by the Reviewer: "...Benchmark performances of 'DeepPBS' (which performs both Groove Readout and Shape Readout modes combined) ..."

5. Figure S7h: What are the red spheres?

AUTHORS: Those are sulfate ions present in the PDB structure (oxygens in red). These are not considered as input in the model. We have updated Fig S7 and S8 removing such ions and small molecules.

COMMENT ON THE CODE: For this review, difficulties were encountered in executing the codes. For review purposes, attempts were made to install DeepPBS on a fresh linux (Ubuntu 22.04) installation, following the instructions provided on the provided github link. Installation of Anaconda and various dependencies went fine but executing the DeepPBS code resulted in crashes related to the PyTorch dependencies, ending either in non-recognition of the GPU (under CUDA 12.2), or inability to proceed in a CPU-only setup. This was not resolved by tinkering version numbers with conda updates and in the environmental (.yml) file. Likely conflicts between evolving versions of PyTorch and CUDA are responsible, and should be addressable by the authors. Nevertheless, for the purpose of the present review, the code did not run as instructed. I might suggest that the authors incorporate some provisions in the package (or at least explicit instructions) to help future-proof the code for users who could not be expected to have access to legacy versions of conda and the various dependencies.

AUTHORS: We thank the Code Reviewer for sharing their experience on attempting to install and execute the code. We agree that the issue was a disagreement between pytorch version and CUDA version. We did test our code for CUDA 11.3 and 11.6 which were the only versions provided by our institution (as mentioned in our GitHub). CUDA 12.2 was unavailable to us at this moment. However, we have now set up a Code Ocean capsule (with provisional DOI 10.24433/CO.0545023.v1, currently inactive but according to our communication the Code Ocean team has sent a peer review link to the journal) which can run the DeepPBS code and we set up a reproducible run for the structure with PDB ID 5X6G (Fig. S7g). The directory structure (under /code/ folder in the capsule) remains analogous to the instructions on GitHub. We hope this effort helps the community to access DeepPBS much more easily.

Additional minor changes:

- For consistency, we updated all figures to use MAE (as opposed to negative MAE, same for RMSE) i.e., all plots now follow “The lower the better” principle.
- A small numerical instability was fixed in the plotting code for Fig. S7, S8, and S9 resulting in the updated PWMs reflecting very minor changes from the previous version. All associated conclusions remain unchanged.

We thank all Reviewers for their very careful evaluations and helpful suggestions, which have improved our manuscript and made it better accessible to a broader audience.

Decision Letter, first revision:

Dear Remo,

Thank you for submitting your revised manuscript "Geometric deep learning of protein–DNA binding specificity" (N METH-A52876B). It has now been seen by the original referees and their comments are below. The reviewers find that the paper has improved in revision, and therefore we'll be happy in principle to publish it in Nature Methods, pending minor revisions to satisfy the referees' final requests and to comply with our editorial and formatting guidelines.

TRANSPARENT PEER REVIEW

Please note: we allow redactions to authors' rebuttal and reviewer comments in the interest of confidentiality. If you are concerned about the release of confidential data, please let us know specifically what information you would like to have removed. Please note that we cannot incorporate redactions for any other reasons. Reviewer names will be published in the peer review files if the reviewer signed the comments to authors, or if reviewers explicitly agree to release their name. For more information, please refer to our FAQ page.

ORCID

Sincerely,
Arunima

Arunima Singh, Ph.D.
Senior Editor
Nature Methods

Reviewer #4 (Remarks to the Author):

I am satisfied with the revisions made by the authors in response to my queries with respect to the manuscript. I am still unable to run the code, and the Code Ocean capsule supplied by the authors is inaccessible to me.

Reviewer #4 (Remarks on code availability):

I am still unable to run the code, and the Code Ocean capsule supplied by the authors is inaccessible to me.

"Editor's note: We followed up with the reviewer regarding code accessibility. The reviewer was able to run the CodeOcean capsule and reported that it was reproducible using the input supplied by the authors. However, the reviewer also had this feedback to share: "It does not mitigate the versioning requirements and conflicts that did not allow me to run the code locally. The ideal solution would be for the authors to provide a (e.g., web-based) front-end that would relieve the end-user from the headaches of the local installation." If you would like to further discuss this, please send me an email/I would be happy to chat with you."

Author Rebuttal, first revision:

RESPONSE TO REVIEWERS

Our ref: NMETH-A52876B

22nd Apr 2024

Dear Dr. Rohs,

Thank you for your patience as we've prepared the guidelines for final submission of your Nature Methods manuscript, "Geometric deep learning of protein–DNA binding specificity" (NMETH-A52876B). Please carefully follow the step-by-step instructions provided in the attached file, and add a response in each row of the table to indicate the changes that you have made. Ensuring that each point is addressed will help to ensure that your revised manuscript can be swiftly handed over to our production team.

Thank you for inviting the submission of the final version of our manuscript. We developed a webserver <https://deeppbs.usc.edu> in response to the comments of the Reviewer and include the webserver in the final version of the manuscript.

Reviewer #4 (Remarks to the Author):

I am satisfied with the revisions made by the authors in response to my queries with respect to the manuscript.

AUTHORS: We are thankful to the Reviewer for their positive assessment of our revised manuscript.

I am still unable to run the code, and the Code Ocean capsule supplied by the authors is inaccessible to me.

Reviewer #4 (Remarks on code availability):

I am still unable to run the code, and the Code Ocean capsule supplied by the authors is inaccessible to me.

"Editor's note: We followed up with the reviewer regarding code accessibility. The reviewer was able to run the CodeOcean capsule and reported that it was reproducible using the input supplied by the authors. However, the reviewer also had this feedback to share: "It does not mitigate the versioning requirements and conflicts that did not allow me to run the code locally. The ideal solution would be for the authors to provide a (e.g., web-based) front-end that would

relieve the end-user from the headaches of the local installation." If you would like to further discuss this, please send me an email/I would be happy to chat with you."

AUTHORS: Based on the suggestion of the Reviewer we have implemented a web-based implementation for DeepPBS available at <https://deeppbs.usc.edu>. We sincerely hope this effort improves accessibility of the method.

Other changes, which all improved the manuscript and did not change the conclusions:

We updated the bHLH example in Fig. 3b to use a protein with deeper sequencing coverage of the HT-SELEX experiment compared to the previously used example. The sequence coverage is now approximately 10-fold higher. The previously used example did not really have enough sequence coverage, which was an oversight.

We updated the p53 case study to now reflect the biologically relevant tetramer binding of the protein to its biological response element. Previously, we showed p53 dimer binding to DNA.

We improved the shape readout component by using mean-padding (instead of 0 padding) to offset inter-base pair features. This resulted in a slight improvement in performance. The text, figures, code repositories have all been adjusted accordingly. All statistical conclusions remain the same.

We thank all Reviewers for their valuable input in the review process and feedback on the work, which has improved our method and manuscript.

Final Decision Letter:

Dear Remo,

I am pleased to inform you that your Article, "Geometric deep learning of protein-DNA binding specificity", has now been accepted for publication in Nature Methods. The received and accepted dates will be August 13, 2023 and June 14, 2024. This note is intended to let you know what to expect from us over the next month or so, and to let you know where to address any further questions.

Over the next few weeks, your paper will be copyedited to ensure that it conforms to Nature Methods style. Once your paper is typeset, you will receive an email with a link to choose the appropriate publishing options for your paper and our Author Services team will be in touch regarding any additional information that may be required. It is extremely important that you let us know now whether you will be difficult to contact over the next month. If this is the case, we ask that you send us the contact information (email, phone and fax) of someone who will be able to check the proofs and deal with any last-minute problems.

Please note that *Nature Methods* is a Transformative Journal (TJ). Authors may publish their research with us through the traditional subscription access route or make their paper immediately open access through payment of an article-processing charge (APC). Authors will not be required to make a final decision about access to their article until it has been accepted. Find out more about Transformative Journals

You may wish to make your media relations office aware of your accepted publication, in case they consider it appropriate to organize some internal or external publicity. Once your paper has been scheduled you will receive an email confirming the publication details. This is normally 3-4 working days in advance of publication. If you need additional notice of the date and time of publication, please let the production team know when you receive the proof of your article to ensure there is

sufficient time to coordinate. Further information on our embargo policies can be found here:
<https://www.nature.com/authors/policies/embargo.html>

If you are active on Twitter/X, please e-mail me your and your coauthors' handles so that we may tag you when the paper is published.

Best regards,
Arunima

Arunima Singh, Ph.D.
Senior Editor
Nature Methods